# From Causal to Concept-Based Representation Learning

**Goutham Rajendran**[1][*]  **Simon Buchholz**[2][*]

**Bryon Aragam**[3]  **Bernhard Schölkopf**[2,4]  **Pradeep Ravikumar**[1]

[1]Machine Learning Dept., Carnegie Mellon University, Pittsburgh, USA
[2]Max Planck Institute for Intelligent Systems, Tübingen, Germany
[3]University of Chicago, Chicago, USA
[4] ELLIS Institute, Tübingen, Germany

## Abstract

To build intelligent machine learning systems, modern representation learning attempts to recover latent generative factors from data, such as in causal representation learning. A key question in this growing field is to provide rigorous conditions under which latent factors can be identified and thus, potentially learned. Motivated by extensive empirical literature on linear representations and concept learning, we propose to relax causal notions with a geometric notion of concepts. We formally define a notion of concepts and show rigorously that they can be provably recovered from diverse data. Instead of imposing assumptions on the "true" generative latent space, we assume that concepts can be represented linearly in this latent space. The tradeoff is that instead of identifying the "true" generative factors, we identify a subset of desired human-interpretable concepts that are relevant for a given application. Experiments on synthetic data, multimodal CLIP models and large language models supplement our results and show the utility of our approach. In this way, we provide a foundation for moving from causal representations to interpretable, concept-based representations by bringing together ideas from these two neighboring disciplines.

## 1  Introduction

A key goal of modern machine learning is to learn representations of complex data that are human-interpretable and can be controlled. Although existing models are known to extract features that are useful and intuitive to humans (e.g. color, shape, size), the sensitivity of these models in capturing these features is notorious [10, 25, 71]: The enormous capacity of contemporary models means that it is possible to reconstruct the input data with meaningless representations (e.g. posterior collapse [39, 24, 129]). Accordingly, understanding how and when useful features can be captured, along with providing assurances on the quality of learned representations, is an ongoing endeavor [27].

A natural approach to this problem is to model the input data $X = (X_1, \ldots, X_{d_x})$ as $X = f(Z)$, where $f$ is a nonlinear transformation that maps structured underlying latent generative factors $Z = (Z_1, \ldots, Z_{d_z})$ to $X$, and then to attempt to recover the model parameters $Z, f$ from $X$. This is an appealing approach since it implies no restrictions on the data $X$, and has the interpretation of recovering "ground truth" factors that generated the data. It is well-known that without additional

---

[*]Equal Contribution

38th Conference on Neural Information Processing Systems (NeurIPS 2024).

assumptions, this is impossible [46, 71], a fact which has led to a long line of work on nonlinear ICA [21, 45] and unsupervised disentanglement [9, 88, 62]. One approach to resolve this limitation is to assume that $Z$ has an intrinsic causal interpretation. This is known as Causal Representation Learning (CRL) [102, 101], which endeavors to identify useful representations through the lens of causality. Structurally, the latent factors $Z$ are assumed to have causal relationships among them, which enables us to reason about effects of interventions and conditioning on these latent factors. CRL studies this setting via an intricate interplay of ideas from causality, latent variable modeling and deep learning, with the main goal being to reconstruct the mixing function $f$ and $Z$, the true generative factors of data, by leveraging causal assumptions. Recent years have witnessed a surge of rigorous results on provably learning causal representations under different assumptions [53, 33, 70, 60, 78, 142, 36, 123, 49, 115]. For example, as long as we have access to interventions on each latent variable $Z_j$ (a total of at least $d_z$ interventions), under weak assumptions on $Z$ and/or $f$, the causal model over $Z$ as well as the model parameters $(Z, f)$ can be uniquely identified [111, 14].

While causal features are intrinsically desirable in many applications, the assumption that we can feasibly perform $\Omega(d_z)$ interventions merits relaxing: Indeed, in complex models, the number of true generative factors $d_z = \dim(Z)$ might be intractably large (e.g. consider all of the latent factors that could be used to describe natural images, video, or text). At the same time, there are yet many other applications where the strict notion of causality may not be needed, and moreover it may not be necessary to learn the *full* causal model over every causal factor. Is there a middle ground where we can simultaneously identify a smaller set of interpretable latent representations, without the need for a huge number of interventions?

In this paper, we study this problem in detail and provide an alternative setting under which latent representations can be provably recovered. Instead of recovering a (potentially huge) number of causal latents with interventions, we settle for recovering a smaller number of *interpretable* concepts without strict interventions. The basic idea is to recover *projections $AZ$* of the generative factors $Z$ that correspond to meaningful, human-interpretable concepts through *conditioning* instead of intervention. The idea to model concepts as linear projections of the generative factors is derived from a growing body of literature (e.g. [90, 55, 130, 77, 5, 22, 30, 17, 118, 81, 38, 75, 103], see Section 3 for even more references) showing that the embeddings learned by modern, high-performant foundation models are not inherently interpretable, and instead capture interpretable concepts as linear projections of the (*apriori*) unintelligible embeddings. While this approach sacrifices causal semantics, it makes up for this with two crucial advantages: 1) Instead of strict interventions in the latent space, it suffices to *condition* on the concepts, and 2) When there are $n$ concepts of interest to be learned, only $n + 2 \ll d_z$ such concept conditionals are needed.

We validate and utilize our theoretical ideas via experiments. First, we validate these theoretical insights on synthetic data, where we use a contrastive algorithm to learn such representations for a given collection of concepts. Moving ahead to real-world data, we probe our theory on embeddings learned by multimodal CLIP models [92]. The training scheme for CLIP aligns with our theoretical setting and therefore, it's reasonable to ask whether they satisfy our observations. Indeed, we show that the concepts in the 3d-Shapes dataset approximately lie in hyperplanes, further supporting our theoretical results. Lastly, we show an effective application of our framework to alignment of large language models (LLMs) where we extend the alignment technique of [66] to make LLMs more truthful.

**Contributions**  In summary, our contributions are:

1. We formalize the notion of distributions induced by abstract concepts in complex domains such as images or text (see Secion 2 for an overview and Section 4.2 for formal definitions). Our definition of concept conditional distributions allows both continuous and fuzzy concepts.

2. We prove near-optimal identifiability results for learning a collection of concepts from a diverse set of environments in Theorem 1. Thus our work can be interpreted as a new direction for identifiable representation learning in order to study when interpretable concepts can be recovered from data.

3. We then verify our guarantees via a contrastive learning algorithm on synthetic data. In addition in Section 6, we support our geometric definition of concepts and our identifiability

result by analysing image embeddings of CLIP-models and we utilize our ideas to improve alignment of LLMs to make them more truthful.

## 2 Overview

In this section, we describe our approach and put it in context of prior developments.

**Defining concepts geometrically** Our starting point is a geometric notion that concepts live in linear directions in neural representation space, known as linearity of representations (see extensive references in Section 3). To make this precise we assume that for observed data $X$ that has an underlying representation $Z$ with $X = f(Z)$ where the latent variables $Z$ follow an arbitrary distribution and $f$ is a (potentially complicated) nonlinear underlying mixing map. We do not assume that $f$ and $Z$ correspond to a ground truth model or that the latent variables $Z$ themselves are related to a causal model or are interpretable and instead only assume linearity of representations (well supported by prior works). In agreement with this hypothesis we define concepts as affine subspaces $AZ = b$ of the latent space of $Z$s, i.e., to a concept $C$ we assign an affine hyperplane $H_C = \{Z \in \mathbb{R}^{d_z} : AZ = b\}$ in the embedding space and we say that $X = f(Z)$ satisfies a concept $C$ if $Z \in H_C$. In this setting the usual goal of CRL to reconstruct $f$ and $Z$ seems to be unneccesary for many applications. Instead we focus on the modest goal of identifying only a (small) set of *concepts we care about*, i.e., we want to be able to decide whether a datapoint $X$ satisfies a concept $C$. Our main result shows that it is possible to identify $n$ concepts given access to $n + 2$ concept conditional distributions. We now compare natural assumptions on type of data for causal representation learning and the setting considered here.

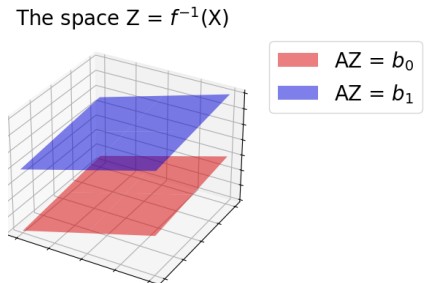

Figure 1: Concepts live in affine subspaces. The two subspaces in the figure correspond to the same concept but of different valuations.

**From interventions to conditioning** It is worth contrasting here the difference between viewing a concept as a generic latent generative factor $Z_i$ that non-linearly mixes together with other latent factors to yield the inputs $X$, versus the geometric notion above, as specifying a linear subspace. In the former, the natural way to provide supervision, i.e. define concept distributions, is to simply intervene on a specific factor $Z_i$ and set it to a particular value (see Section 3 for references).In the latter however, it is most natural to condition on the concept, i.e., $Z \in H$.

This shift is aligned with the growing interest to relax the notion of interventions, and consequently dilute the notion of causality [15, 100, 4], although it is still open how to properly achieve this. Two key drivers of this trend are as follows. The first is that the number of additional datasets required is $d_z$ [46, 71, 53, 14], which is infeasible in many settings [2]. The second is that the various assumptions that go into these works are often difficult to achieve, such as requiring perfect interventions [111, 14]. Compared to interventional data, *conditional* data is often easier to acquire, obtained by conditioning on particular values of the latent factors (see also Appendix B.2).

**Concept conditional distributions** We now formalize conditioning on a concept. The obvious approach to define concept conditional distributions is to simply condition on $Z \in H_C$, so $p_C(Z) = p(Z|Z \in H_C)$ where $p$ is a base distribution of $Z$ on $\mathbb{R}^{d_z}$. However, this suffers from the drawbacks that it is mathematically subtle to condition on sets of measure 0 and this does not account for inherent noise in the learned representations. Therefore we relax this strict conditioning by drawing inspiration from how data is collected in practice: We sample $X$ from the base distribution and then keep it if it satisfies our concept $C$. This leads us to define $p_C(Z) \propto p(Z)q(Z|C)$ where $q$ is defined to be the probability that $Z$ is *perceived* to be in $H$ by the data collector and can be chosen to incorporate noise in our data gathering scheme. Therefore, this can also be viewed from a Bayesian information gathering viewpoint, as well as a stochastic filter standpoint. This is the notion we study in this work

---

[2]Exceptions are [57, 42], which use clever inductive biases to limit the number of environments needed.

(Definition 3) and we develop theoretical techniques to guarantee identifiability in this formulation. Depending on the specific setting other types of conditional distributions might be utilized to describe the available data and we discuss some options in Appendix C.

**Connection to XAI**   One important class of applications is human-in-the-loop (HIL) settings, where the hope is to recover human interpretable concepts.   Apriori, there is no reason why the high-dimensional underlying latent $Z$s should be interpretable, particularly for large $d_z$.   To make the situation worse, there may be other potentially composite concepts that humans can conjure up; indeed, there is no limit to the number of concepts that humans can conjure, and these almost certainly are not going to correspond to the true $Z$s. Because of this mismatch between the "true" $Z$s and interpretable concepts, even if we could recover the $Z$s, they may not be interpretable to humans and therefore less useful for downstream HIL purposes. The second caveat, and which follows from the first as a soft corollary, is that interventions on these will be much harder to obtain, particularly perfect interventions which require us to isolate the generative mechanisms of the latent factor with the other factors. Furthermore, some concepts are not manipulable, e.g. psychological traits, biological mechanisms, so obtaining interventional data is not possible in these applications.   Accordingly, there has been considerable recent work, especially in the explainable AI (XAI) community on extracting human-interpretable concepts from latent generative factors, or more generally from complex foundation model representations [55, 103, 18, 41, 137]. To do so in an identifiable and automated way with as little supervision as possible is a significant open problem.

## 3   Related work

**Causal representation learning and concept discovery**   Causal representation learning (CRL) [102, 101] aims to learn generative factors of high-dimensional data. This exciting field has seen significant progress in the last few years [53, 11, 106, 60, 78, 57, 114, 14, 36, 1, 127, 63].   A fundamental perspective in this field is to ensure that the model parameters we attempt to recover are identifiable [53, 25, 129]. We will elaborate more on the connection of our framework to CRL in Appendix B. Concept discovery is an important sub-field of machine learning which extracts human-intepretable concepts from pre-trained models. We do not attempt to list the numerous works in this direction, see e.g., [103, 18, 135, 74, 82, 99, 58, 104, 89, 114]. However, theoretical progress in this direction is relatively limited. The work [63] studies when concepts can be identified provided the non-linear model is known in advance, whereas we show concept identifiability for unknown non-linearity, while simultaneously allowing entangled concepts. Prior works have also attempted to formalize the notion of concepts [130, 85, 103, 61], however their definitions seem specific to the model and domain under consideration, e.g., [85, 52] focus on binary concepts via large language model representations of counterfactual word pairs, whereas our general concept definitions are applicable to all domains.

**Linearity of representations**   Sometimes referred to as the linear representation hypothesis, it is commonly believed that well-trained foundation models in multiple domains learn linear representations of human-interpretable concepts, with experimental evidence going back at least a decade [77, 113, 5]. This has been experimentally observed in computer vision models [90, 94, 8, 31, 55, 19, 130, 120], language models [77, 87, 5, 22, 117, 30], large language models [17, 118, 81, 79, 66, 85, 38, 52], and other intelligent systems [75, 103]. Various works have also attempted to justify why this happens [64, 5, 35, 3, 32, 105]. We take a different angle: Given that this phenomenon has been observed for certain concepts of interest, how does this enable recovery of the concepts themselves? Consequently, our model assumptions are well-founded and our theory applies to multiple domains of wide interest.

## 4   Setup

In this section, we provide a formal definition of concepts, which are high-level abstractions present in data. This allows us to develop a theoretical framework for associated data distributions and identifiability theory. For the sake of intuition, we can think of the data as images of different objects and the color of the object as a concept.

## 4.1 Generative model

We assume that the observed data $X$ lies in a space $\mathcal{X} \subseteq \mathbb{R}^{d_x}$ of dimension $d_x$ and has an underlying representation $X = f(Z)$ for latent variables $Z$ that lie in a latent concept space $\mathbb{R}^{d_z}$ of dimension $d_z$. In contrast to most prior works we do not necessarily assume that $Z$ represents the true underlying mechanism that generated the data. Instead we simply assume that the latent representation has the geometric property that it maps certain regions of the observation space to linear subspaces of the latent space (motivated by previous work; see Section 3). Our first assumption is standard:

**Assumption 1** (Mixing function). *The non-linear $f$ is injective and differentiable.*

We make no additional assumptions on $f$: The map from $Z \to X$ can be arbitrarily non-linear.

We now define concepts living in the latent space $\mathbb{R}^{d_z}$. Before presenting the general definition of multidimensional concepts, we outline the basic ideas in the simplified setting of a one-dimensional concept. Consider the color "red" as a concept. Different images have different levels of "redness" in them, so this concept is measured on a continuous scale, represented by a valuation $b \in \mathbb{R}$. An (atomic) concept is then represented by a vector $a \in \mathbb{R}^{d_z}$ such that $\langle a, Z \rangle = \langle a, f^{-1}(X) \rangle$ encodes the "value" of the concept in $X$, as measured in the latent space. More precisely, for a given valuation $b \in \mathbb{R}$, the set of all observations $X$ that satisfy this concept is given by $\{X = f(Z) | \langle a, Z \rangle = b\}$. For instance, for an object in an image $X$, if $a \in \mathbb{R}^{d_z}$ is the concept of red color, $b \in \mathbb{R}$ could indicate the intensity; then all datapoints $X$ satisfying this concept, i.e., all images with an object that has color red with intensity $b$, can be characterized as $X = f(Z)$ where $Z$ satisfies $\langle a, Z \rangle = b$. For a 3D visualization, see Fig. 1 We make this intuition formal below.

**Definition 1** (Concepts). *A concept $C$ is a linear transformation $A : \mathbb{R}^{d_z} \to \mathbb{R}^{d_C}$. The dimension of the concept will be denoted by $\dim(C) = d_C$. A valuation is a vector $b \in \mathbb{R}^{d_C}$ and we say that a datapoint $X$ satisfies the concept $C$ with valuation $b$ if $AZ = b$ where $Z = f^{-1}(X)$.*

In this work, we are interested in learning a collection of $m$ concepts $C^1, \ldots, C^m$ from observed data. By left multiplying by the pseudo-inverse $A^+$, we can equivalently assume $A$ is a projector matrix. However, the current definition is more suitable for embeddings of real models.

When we talk of learning concepts $C$, we are in particular interested in learning the evaluation map $Af^{-1}(x)$. This is a more modest objective than learning the entire map $f$ which is the usual goal in, say, CRL. While the latter typically requires stringent assumptions, in particular $\Omega(d_z)$ environments are necessary, our weaker identifiability results only need $O(d_C) \ll O(d_z)$ environments. To simplify our analysis, we make use of the following definition:

**Definition 2** (Atoms). *An atom (short for atomic concept) is any concept $C$ with $\dim(C) = 1$.*

The idea is that we can view each concept as being composed of atomic concepts in the following sense: Atomic concepts are fundamental concepts that live in a space of co-dimension 1 in latent space, and thus are equivalently defined by vectors $a \in \mathbb{R}^{d_z}$. For example, concepts such red color, size of object, etc., may be atomic concepts. Any generic concept is then composed of a collection of atomic concepts, e.g., the concept $C$ of all small dark red objects will correspond to $\dim(C) = 2$ with row 1 corresponding to the atomic concept of red color with large valuation (dark red objects) and row 2 corresponding to the atomic concept of object size with low valuation (small objects).

## 4.2 Data distributions

We now define the distributions of datasets over concepts. We will predominantly work with distributions of $Z$ over $\mathbb{R}^{d_z}$, as the resulting distribution of $X = f(Z)$ over $\mathbb{R}^{d_x}$ can be obtained via a simple change of variables.

To build intuition, consider the case where we first collect a base dataset with some underlying distribution and then collect concept datasets via filtering. For instance, we could first collect a set of images of all objects and then, to collect a dataset of dark red colored objects, we filter them to only keep images of dark red colored objects. We call the former the *base distribution* and the latter the *concept conditional distribution* corresponding to our concept.

Fix a nonlinearity $f$. We assume that the base data distribution is the distribution of $X = f(Z)$ with $Z \sim p$, where $p$ is the underlying distribution on $\mathbb{R}^{d_z}$. In what follows, we will abuse notation and use $p$ for both the distribution and the corresponding probability density which we assume exists. In

contrast to existing work that puts assumptions on $p$ (such as linearity, Gaussianity, etc.), we make no assumptions on $p$ since we wish to model real-life datasets which could be very arbitrary.

We now define the concept conditional distribution, which is a distribution over $X$ that is induced by noisy observations of a particular concept at a particular valuation. Formally, assume we want to condition on some atomic concept $a \in \mathbb{R}^{d_z}$ with valuation $b$. It is reasonable to assume that this conditioning is a noisy operation. For instance, humans are great at distilling concepts from noisy images, e.g., they recognize cars in a misty environment. We formalize this by assuming that data collection is based on a noisy estimate $\widetilde{b} = \langle a, z \rangle + \epsilon$ where $\epsilon$ is independent of $z$ and its density is a symmetric distribution with density $q(\epsilon)$. Then we consider the distribution

$$
\begin{aligned}
p_C(z) = p(z|\widetilde{b} = b) &\propto p(\widetilde{b} = b|z)p(z) \\
&= q(b - \langle a, z \rangle)p(z)
\end{aligned}
\tag{1}
$$

where we used Bayes theorem in the last step. This definition directly extends to higher dimensional concepts which are concisely defined as follows.

**Definition 3** (Concept conditional distribution). *For a concept $C$ with associated linear map $A$ and an arbitrary valuation $b \in \mathbb{R}^{dim(C)}$, we define the concept conditional distribution to be the set of observations $X$ respecting this concept, which is defined as the distribution of $X = f(Z)$ where $Z \sim p_C$ with*

$$
p_C(Z) \propto p(Z) \prod_{k=1}^{\dim(C)} q((AZ - b)_k).
\tag{2}
$$

This is by no means the only possible definition, and we present feasible alternate definitions in Appendix C. We remark that our formulation is related to the iVAE setting [53] and the auxiliary variable setting for identifiable ICA in Hyvarinen et al. [48] and we discuss the relation later. The majority of recent identifiability results relied on interventional data while we only consider conditional information here.

### 4.3 Concept learning and identifiability

We are ready to define our main problem of interest.

**Problem 1.** *We are given an observational dataset $X^0 = f(Z^0)$ corresponding to the latent base distribution $p$ along with datasets $X^1, \ldots, X^m$ from $m$ environments corresponding to concept conditional datasets for different concepts $C^1, \ldots, C^m$ and corresponding valuations $b^1, \ldots, b^m$ over the same latent space $\mathbb{R}^{d_z}$ with the same mixing $f$. Under what conditions (and up to which symmetries) can we learn the concepts $C^1, \ldots, C^m$, which includes the linear maps $A^1, \ldots, A^m$, and the concept valuations $A^1 f^{-1}(x), \ldots, A^m f^{-1}(x)$?*

Toward this end, a fundamental question is whether this problem is even possible, i.e., whether it is well-defined. This is known as the question of identifiability [53, 25, 129, 57, 43]. Therefore, we make the following definition. Informally, for the setting above, we say that the concepts $(C^1, A^1), \ldots, (C^m, A^m)$ with associated nonlinearity $f$ are identifiable (and thus learnable) if for any other collection of different parameters that fit the data, they are linearly related to the true parameters.

**Definition 4** (Identifiability). *Given datasets $X^0, X^1, \ldots, X^m$ corresponding to the observational distribution and $m$ concepts $C^1, \ldots, C^m$ with underlying latent base distribution $p$ on $\mathbb{R}^{d_z}$, nonlinearity $f$, linear maps $A^1, \ldots, A^m$ and valuations $b^1, \ldots, b^m$, we say the concepts are identifiable if the following holds: Consider any different collection of parameters $\widetilde{f}, \widetilde{d_z}, \widetilde{p}$, concepts $(\widetilde{C^1}, \widetilde{A^1}), \ldots, (\widetilde{C^m}, \widetilde{A^m})$ and valuations $\widetilde{b^1}, \ldots, \widetilde{b^m}$ that also generate the same observations $X^0, X^1, \ldots, X^m$. Then there exists a shift $w \in \mathbb{R}^{d_z}$, permutation matrices $P^e$ and invertible diagonal matrices $\Lambda^e$ such that for all $e$ and $x$,*

$$
\widetilde{A}^e \widetilde{f}^{-1}(x) = \Lambda^e P^e A^e (f^{-1}(x) + w),
\tag{3}
$$

*i.e., we can evaluate the concept evaluations on the data up to linear reparametrizations. Moreover, there exists a linear map $T : \mathbb{R}^{\widetilde{d_z}} \to \mathbb{R}^{d_z}$ such that the concepts and their evaluations satisfy*

$$
\widetilde{A}^e = P^e A^e T^{-1}, \quad \widetilde{b}^e = \Lambda^e P^e (b^e - A^e w).
\tag{4}
$$

Identifiability implies we can identify the nonlinear map $f^{-1}$ within the span of the subspace of the concepts of interest, and therefore we can recover the concepts of interest from our data. That is, if certain concepts are identifiable, then we will be able to learn these concept representations up to linearity, even if they can be highly nonlinear functions of our data. Such concept discovery is useful because they can then be used for further downstream tasks such as controllable generative modeling.

We emphasize that in contrast to previous work we are not aiming to identify $f$ completely and indeed, no stronger identifiability results on $f$ can be expected. First, we cannot hope to resolve the linear transformation ambiguity because the latent space is not directly observed. In other words, a concept evaluation can be defined either as $\langle a, Z \rangle$ or as $\langle Ta, T^{-\top} Z \rangle$ for an invertible linear map $T$. For the purposes of downstream tasks, however, this is fine since the learned concepts will still be the same. Second, we cannot expect to recover $f^{-1}$ outside the span of the concepts because we do not manipulate the linear spaces outside the span therefore we do not learn this information from our observed data so this is also tight. The permutation matrix captures the fact that the ordering of the concepts does not matter. Therefore, this definition captures the most general identifiability guarantee that we can hope for in our setting and furthermore, this suffices for downstream tasks such as controllable data generation.

Because we will only be interested in recovering the set of concepts up to linear transformations, without loss of generality, we will fix the base collection of atomic concepts. That is, we assume that each concept $C^e$ (where $1 \leq e \leq m$ indexes the environment) corresponds to a linear map $A^e$ whose rows are a subset of $\mathcal{C}$, where $\mathcal{C} = \{a_1, \dots, a_n\}$ is a set of atomic concepts that we wish to learn. Moreover, we assume that they are linearly independent, since we want them to encode distinct concepts. This is formalized as follows.

**Assumption 2.** *There exists a set of atomic concepts $\mathcal{C} = \{a_1, \dots, a_n\}$ of linearly independent vectors such that for each concept $C^e$ under consideration the rows of the concept matrix $A^e$ are contained in $\mathcal{C}$, i.e., $(A^e)^t e_i \in \mathcal{C}$. We denote the indices of the subset of $\mathcal{C}$ that appear as rows of $A^e$ by $S^e$ and we assume that all concepts in $\mathcal{C}$ appear in some environment $e$ (where an environment corresponds to a concept conditional distribution), i.e., $\bigcup_e S^e = [n]$.*

**Remark 1.** *Definition 4 implies that the atoms can be identified in the sense that there is a permutation $\pi \in \mathfrak{S}_n$ and $\lambda_i \neq 0$ such that for $T$ as in Definition 4 and some $\lambda_i$*

$$\widetilde{a}_{\pi(i)}^{\top} = a_i^{\top} T^{-1} \tag{5}$$

$$\langle \widetilde{a}_{\pi(i)}, \widetilde{f}^{-1}(x) \rangle = \lambda_i \left( \langle a_i, f^{-1}(x) \rangle + \langle a_i, w \rangle \right), \tag{6}$$

*i.e., we can evaluate the valuations of the atomic concepts up to linear reparametrization.*

## 5 Main Result

In this section, we present our main result on identifying concepts from data. The punchline is that when we have rich datasets, i.e., sufficiently rich concept conditional datasets, then we can recover the concepts. Crucially, we only require a number of datasets that depends only on the number of atoms $n$ we wish to learn (in fact, $O(n)$ datasets), and not on the underlying latent dimension $d_z$ of the true generative process. This is a significant departure from many existing works, since the true underlying generative process could have $d_z = 1000$, say, whereas we may be interested to learn only $n = 5$ concepts, say. In this case, approaches based on CRL necessitate at least $\sim 1000$ *interventional* datasets, whereas we show that $\sim n + 2 = 7$ *conditional* datasets are enough if we only want to learn the $n$ atomic concepts. We will explain the connection to CRL in Appendix B. Let us now discuss our main assumptions.

**Assumption 3.** *The noise distribution $q$ is Gaussian, i.e. $q \sim N(0, \sigma^2)$ for some $\sigma^2 > 0$.*

We choose Gaussian noise since it is a conventional modeling choice. However, it would be feasible to consider other noise families and we expect similar results to hold (albeit with modified proof techniques). We now relate the concepts $C^e$ to the atoms. Recall that we defined the index sets $S^e = \{i \in [n] : a_i \in \mathcal{C} \text{ is a row of } A^e\}$ of atomic concepts in environment $e$.

We define the environment-concept matrix $M \in \mathbb{R}^{m \times n}$ indexed by environments and atoms by

$$M_{ei} = \begin{cases} \frac{1}{\sigma^2} & \text{if } i \in S^e \\ 0 & \text{otherwise.} \end{cases} \tag{7}$$

Similarly, we consider the environment-valuation matrix $B \in \mathbb{R}^{m \times n}$ given by

$$B_{ei} = \begin{cases} \frac{b_k^e}{\sigma^2} & \text{if } i \in S^e \text{ and row } k \text{ of } A^e \text{ is } a_i, \\ 0 & \text{otherwise.} \end{cases} \tag{8}$$

Our first assumption ensures that the concept conditional distributions are sufficiently diverse.

**Assumption 4** (Environment diversity I). *The environment-concept matrix $M \in \mathbb{R}^{m \times n}$ has rank $n$ and there is a vector $v \in \mathbb{R}^m$ such that $v^\top M = 0$ and all entries of $v^\top B$ are non-zero where $B$ denotes the environment-valuation matrix.*

We remark that this assumption can only hold for $m \geq n + 1$ and indeed is satisfied under mild assumptions on the environments if $m = n + 1$, as the following lemma shows. Note that the condition on $B$ ensures that the concept valuations $b_k^e$ are not equal for all environments $e$ which would prevent identifiability.

**Lemma 1.** *Assumption 4 is satisfied almost-surely if there are $n + 1$ concept conditional distributions such that every $n$ rows of the environment-concept matrix are linearly independent and the $b^e$ are drawn independently according to a continuous distribution.*

We also assume one additional diversity condition. To motivate this, observe if two concepts always occur together, it's information-theoretically impossible to distinguish them, e.g., if an agent only sees red large objects (i.e. all red objects are large and all large objects are red), it will be unable to disambiguate the "red" concept from the "large" concept. Therefore, we make the following assumption.

**Assumption 5** (Environment diversity II). *For every pair of atoms $a_i$ and $a_j$ with $i \neq j$ there is an environment $e$ such that $i \in S^e$ and $j \notin S^e$.*

We remark that these are the only assumptions about the sets $S^e$. In particular, we do not need to know the sets $S^e$. In the proof, we will extract these sets based on a the signatures they leave on the datasets. We can now state our main result.

**Theorem 1.** *Suppose we are given $m$ context conditional datasets $X^1, \ldots, X^m$ and the observational dataset $X^0$ such that Assumptions 1-5 hold. Then the concepts are identifiable as in Definition 4.*

**Remark 2.** *Assumption 4 can only be satisfied for $m \geq n + 1$, i.e., the result requires at least $n + 2$ environments. On the other hand, Lemma 1 assures that $n + 2$ environments are typically sufficient. We expect that the result could be slightly improved by showing identifiability for $n + 1$ environments under suitable assumptions. However, this would probably require more advanced techniques from algebraic statistics [28] compared to the techniques we employ here.*

As mentioned before, our setting somewhat resembles the iVAE setting in Khemakhem et al. [53] and therefore, their proof techniques can also be applied, with several modifications, to derive identifiability results in our setting (however our formulation and application are very different). However, this approach will require more environments because their main assumption is that the matrix $\Lambda = (M, B) \in \mathbb{R}^{m \times 2n}$ has rank $2n$ so that $2n + 1$ environments are necessary. Moreover, this rank condition is much stronger than Assumption 4. For completeness and as a warm-up we prove this result in Appendix A. The full proof of Theorem 1 is fairly involved and is deferred to Appendix A.

## 6 Experiments

In this section, we present experiments to validate and utilize our framework. We first verify our results on synthetic data, via a contrastive learning algorithm for concept learning. Then, we focus on experiments involving real-world settings, in particular on image data using multimodal CLIP models and text data using large language models (LLMs).

**End-to-end Contrastive learning algorithm and Synthetic experiments** We validate our framework on synthetic data as follows. We sample the base distribution from a Gaussian Mixture model and experiment with both linear and nonlinear mixing functions (details deferred to Appendix G). The number of concepts $n$ is intentionally chosen to be less than the ground truth dimension $d_z$ and

the number of concept condtional distributions is $m = n + 1$ as per our theory. Inspired by [14], we use a contrastive learning algorithm to extract the concepts. Here we sketch the key ideas of the algorithm and defer details to Appendix F.

The core idea for the algorithm is as follows. For each concept conditional distribution $X^e$, we train a neural network to distinguish concept samples $x \sim X^e$ from base samples $x \sim X^0$. Under our assumptions the log-odds of these two distributions have the following simple parametric form in terms of the environment-concept matrix and the environment-valuation matrix

$$\ln(p^e(Z)) - \ln(p(Z)) = \sum_{i=1}^{n} \left( -\frac{1}{2} M_{ei} \langle a_i, Z^e \rangle^2 + B_{ei} \langle a_i, Z^e \rangle \right) + c_e \tag{9}$$

for some constants $c_e$ (see Lemma 3 for the precise statement).

Then, to learn the $n$ atomic concepts up to linearity, we build a neural architecture for this classification problem with the final layer mimicking the log-odds expression above, which can then be trained end-to-end. Because of the careful parametrization of the last layer, this will encourage the model to learn the representations as guaranteed by our results. We can assume without loss of generality that the concept vectors we learn are the first coordinate vectors because concepts are only identifiable up to linear transformations (see Definition 4). In other words, we consider an encoder neural network $h^\theta$ with parameters $\theta$ and the valuation of atomic concept $i$ is $h_i^\theta$. Therefore, for each environment $e$, we train classifiers of the form

$$g_e(X, \alpha^e, \beta_k^e, \gamma_k^e, \theta) = \alpha^e - \sum_{k=1}^{n} \beta_e^k (h_k^\theta(X))^2 + \sum_{k=1}^{n} \gamma_e^k h_k^\theta(X) \tag{10}$$

using standard cross-entropy loss, where $\alpha^e, \beta_k^e, \gamma_k^e$ are the parameter of the last layer and $\theta$ parametrizes the decoder.

In Table 1, we report the $R^2$ and Mean Correlation Coefficient (MCC) metrics [53, 54] with respect to the ground truth concept valuations. In addition we provide results for larger values of $d_x$ and $d_z$ in Table 7 in Appendix G. There are no baselines since we are in a novel setting, but our metrics are comparable to and often surpass what's usually reported in such highly nonlinear settings [132, 14]. We remark that variations of the contrastive method can be designed for harder synthetic settings and different problems related to concept discovery. However, we will move onto real-life data experiments next.

Table 1: Linear identifiability when number of concepts $n$ is less than latent dimension $d_z$ with observed dimension $d_x$, averaged over 5 seeds.

| Mixing ($f$) | $(n, d_z, d_x)$ | $R^2\uparrow$ | MCC$\uparrow$ |
|---|---|---|---|
| Linear | (2, 3, 4) | $0.98 \pm 0.01$ | $0.98 \pm 0.03$ |
| Nonlinear | (2, 3, 4) | $0.94 \pm 0.06$ | $0.96 \pm 0.04$ |
| Linear | (3, 4, 6) | $0.99 \pm 0.01$ | $0.86 \pm 0.08$ |
| Nonlinear | (3, 4, 6) | $0.97 \pm 0.03$ | $0.92 \pm 0.07$ |
| Linear | (4, 8, 10) | $0.97 \pm 0.01$ | $0.87 \pm 0.06$ |
| Nonlinear | (4, 8, 10) | $0.94 \pm 0.03$ | $0.87 \pm 0.06$ |

**Probing the theory on multimodal CLIP models** A real world example that approximately matches the setting considered in this paper is the training of the multimodal CLIP models [92]. They are trained by aligning the embeddings of images and their captions. We can view the caption as an indicator of the concepts present in the image. Thus the data provides access to several concept conditional distributions such as the collection of all images having the label 'A dog', but also to more complex distributions consisting of more than one atomic concept such as images labeled 'A red flower'. We embed images from the 3d-Shapes Dataset [16] with known factors of variation into the latent space of two different pretrained CLIP models. Using logistic regression we learn atomic concepts for each of the factors of variations (see Appendix D.1 for details) and then evaluate the concept valuations of the learned atomic concept on held out images. We show the results for the shape attribute in Figure 2 (further results are in Appendix D.2). The results show that there are indeed linear subspaces of the embeddings space that represent certain concepts. Moreover, the learned valuations for different models are approximately linearly related as predicted by Theorem 1. We emphasize that, while these observations highlight the relevance of the theory developed in this paper, they do not explain the behavior of CLIP models, as their training objective does not directly align with our theoretical framework.

**Alignment of LLMs** Finally, we show an application of our framework to interpret representations of LLMs and improve alignment techniques. In particular, we exploit our ideas to improve the

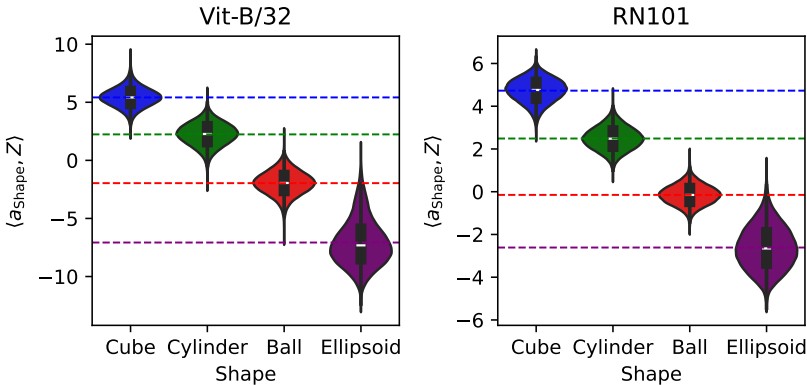

Figure 2: Violin plot of the concept valuations $\langle a_{\mathrm{Shape}}, Z \rangle$ for the different shapes and a vision transformer CLIP embedding (left) and a residual network CLIP embedding (right). Results show concentration of the concept valuations around the concept planes indicated by the horizontal lines.

Inference-Time Intervention technique [66] to promote LLMs to be more truthful, i.e. the downstream task is to take pre-trained LLMs and during inference, change the valuation of the truthfulness concept from *false* to *true*, without affecting any other orthogonal concepts. Motivated by our framework, we propose to replace steering vectors by steering matrices for better alignment. Experiments on LLaMA [119] show an improvement of the TruthfulQA dataset [68] accuracy. Additional details, including a self-contained introduction to large language models (LLMs) and the Inference-Time Intervention (ITI) technique are deferred to Appendix E.

## 7    Conclusion

In this work, we study the problem of extracting concepts from data, inspired by techniques from causal representation learning. This is an approach to bridge identifiability theory in (causal) representation learning and the field of concept-based learning. For this, we geometrically define concepts as linear subspaces, well-supported via extensive empirical literature. With this formal definition of concepts, we study under what conditions they can be provably recovered from data. Our rigorous results show that this is possible under the presence of only conditional data, requiring far fewer distributions than the underlying latent dimension. Finally, synthetic experiments, multimodal CLIP experiments and LLM alignment experiments verify and showcase the utility of our ideas.

**Acknowledgments**    We acknowledge the support of AFRL and DARPA via FA8750-23-2-1015, ONR via N00014-23-1-2368, NSF via IIS-1909816, IIS-1955532, IIS-1956330, and NIH R01GM140467. We also acknowledge the support of the Tübingen AI Center, the Deutsche Forschungsgemeinschaft (DFG, German Research Foundation) under Germany's Excellence Strategy – EXC number 2064/1 – Project number 390727645, and the Robert H. Topel Faculty Research Fund at the University of Chicago Booth School of Business.

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

# A  Proofs of the main results

In this appendix we provide the proofs of our results, in particular the proof of our main result, Theorem 1. However, as a warm-up we first start in Appendix A.1 with a proof of the simpler result that can be shown based on the iVAE approach. In Appendix A.2 we prove Theorem 1 and in Appendix A.3 we prove the additional lemmas that appear in the paper.

## A.1  Proof of identifiability with $2n + 1$ environments

As a warm-up and to provide a connection to earlier results we show here how to obtain identifiability by adapting the iVAE framework to our context. Indeed, our mathematical setting is related to the setting used in [53] in the sense that the environments are generated by modulation with certain exponential families. Therefore, we can essentially apply their proof techniques to prove identifiability (with some modifications), albeit this requires the suboptimal number of $2m + 1$ environments (there are two sufficient statistics for the Gaussian distribution).

**Theorem 2.** *Suppose data satisfies Assumption 1, 2, and 3 and the environment statistics matrix $\Lambda$ has rank $2n$. Assume we know the number of atoms $n$. Then identifiability in the sense of Definition 4 holds.*

We remark that the rank condition can only be satisfied for $2n + 1$ environments (observational distribution and $2n$ concept conditional distributions. For this theorem the assumption that the filtering distribution is always the same is not necessary. Instead we could consider variances $(\sigma_k^e)^2$ depending on environment $e$ and row $k$, i.e., the filtering distribution $q_{(\sigma_k^e)^2}$ is Gaussian with varying variance. The generalization of the environment-concept matrix $M \in \mathbb{R}^{m \times n}$ is given by

$$M_{ei} = \begin{cases} \frac{1}{(\sigma_k^e)^2} & \text{if } i \in S^e \text{ and row } k \text{ of } A^e \text{ is } a_i \\ 0 & \text{otherwise.} \end{cases} \tag{11}$$

Similarly the generalization of the environment-valuation matrix $B \in \mathbb{R}^{m \times n}$ is given by

$$B_{ei} = \begin{cases} \frac{b_k^e}{(\sigma_k^e)^2} & \text{if } i \in S^e \text{ and row } k \text{ of } A^e \text{ is } a_i, \\ 0 & \text{otherwise.} \end{cases} \tag{12}$$

We now prove Theorem 2. We use essentially the same ideas as in the proof of Theorem 1 in Khemakhem et al. [53] (followed by the same reasoning as in Sorrenson et al. [108], Kivva et al. [57] but since our concepts are not axis aligned and we only extract some information about the mixing we give a complete proof.

*Proof of Theorem 2.* Suppose there are 2 sets of parameters that generate the same data $X^0, X^1, \ldots, X^m$. Denote by $\sim$ the latter set of parameters, e.g., $X^e$ is distributed as $\widetilde{f}(\widetilde{Z}^e)$ where $\widetilde{Z}^e \in \mathbb{R}^{\widetilde{d}_z}$ corresponds to the concept class $\widetilde{C}^e$ with distribution $\widetilde{Z}^e \sim \widetilde{p}^e$ and the same distribution is generated by $f(Z^e)$ where $f$ and $\widetilde{f}$ are injective and differentiable. Let $\mathcal{C} = \{a_1, \ldots, a_n\}$ be the set of atomic concepts in the first setting and let $\widetilde{\mathcal{C}} = \{\widetilde{a}_1, \ldots, \widetilde{a}_n\}$ be the set of atomic concepts in the second setting (here we use that $n$ is assumed to be known). We also consider the transition function $\varphi = \widetilde{f}^{-1} f$ and in the following we always write $\widetilde{Z} = \varphi(Z)$. The equality $f(Z^e) \overset{\mathcal{D}}{=} X^e \overset{\mathcal{D}}{=} \widetilde{f}(\widetilde{Z}^e)$ implies $\varphi(Z^e) \overset{\mathcal{D}}{=} \widetilde{Z}^e$. This implies that for all environments $e$

$$p^e(Z) = |\det J_{\varphi^{-1}}| \cdot \widetilde{p}^e(\widetilde{Z}) \tag{13}$$

Taking the logarithm and subtracting this for some $e = 1, \ldots, m$ from the base distribution we obtain

$$\ln(p(Z)) - \ln(p^e(Z)) = \ln(\widetilde{p}(\widetilde{Z})) - \ln(\widetilde{p}^e(\widetilde{Z})). \tag{14}$$

Using the definition (2) we can rewrite for some constants $c_e$ and $c_e'$

$$\begin{aligned} \ln(p(Z)) - \ln(p^e(Z)) &= \sum_{k=1}^{\dim(C_e)} \frac{(A^e Z^e - b^e)_k^2}{2(\sigma_k^e)^2} - c_e' \\ &= \sum_{i=1}^{n} \left( \frac{1}{2} M_{ei} \langle a_i, Z^e \rangle^2 - B_{ei} \langle a_i, Z^e \rangle \right) - c_e. \end{aligned} \tag{15}$$

Here we used the environment-concept matrix and the environment-valuation matrix in the second step which were defined in (7) and (8) (in (11) and (12) for varying variance). We define the vector $\boldsymbol{p}(Z)$ with components $\boldsymbol{p}_e(Z) = \ln(p(Z)) - \ln(p^e(Z))$. Then we find the relation

$$\boldsymbol{p}(Z) = \frac{1}{2}M\begin{pmatrix} \langle a_1, Z\rangle^2 \\ \vdots \\ \langle a_n, Z\rangle^2 \end{pmatrix} - B\begin{pmatrix} \langle a_1, Z\rangle \\ \vdots \\ \langle a_n, Z\rangle \end{pmatrix}. \tag{16}$$

Together with (14) we conclude that

$$\frac{1}{2}M\begin{pmatrix} \langle a_1, Z\rangle^2 \\ \vdots \\ \langle a_n, Z\rangle^2 \end{pmatrix} - B\begin{pmatrix} \langle a_1, Z\rangle \\ \vdots \\ \langle a_n, Z\rangle \end{pmatrix} = \frac{1}{2}\widetilde{M}\begin{pmatrix} \langle \widetilde{a}_1, \widetilde{Z}\rangle^2 \\ \vdots \\ \langle \widetilde{a}_n, \widetilde{Z}\rangle^2 \end{pmatrix} - \widetilde{B}\begin{pmatrix} \langle \widetilde{a}_1, \widetilde{Z}\rangle \\ \vdots \\ \langle \widetilde{a}_n, \widetilde{Z}\rangle \end{pmatrix} \tag{17}$$

Since by assumption $\widetilde{\Lambda} = (\widetilde{M}, \widetilde{B}) \in \mathbb{R}^{m \times 2n}$ has rank $2n$ there is a vector $v$ such that $v^\top \widetilde{M} = 0$ and $v^\top \widetilde{B} = -\boldsymbol{e}_i$ ($\boldsymbol{e}_i \in \mathbb{R}^{d_z}$ denotes the $i$-th standard basis vector). Thus we find that

$$\langle \widetilde{a}_i, \widetilde{Z}\rangle = \frac{1}{2}v^\top M\begin{pmatrix} \langle a_1, Z\rangle^2 \\ \vdots \\ \langle a_n, Z\rangle^2 \end{pmatrix} - v^\top B\begin{pmatrix} \langle a_1, Z\rangle \\ \vdots \\ \langle a_n, Z\rangle \end{pmatrix}. \tag{18}$$

In other words $\langle \widetilde{a}_i, \widetilde{Z}\rangle$ can be expressed as a quadratic polynomial in $Z$. We apply the same reasoning for $\langle \widetilde{a}_i, \widetilde{Z}\rangle^2$, i.e., pick a vector $v'$ such that $\frac{1}{2}v'^\top \widetilde{M} = \boldsymbol{e}_i$ and $v'^\top \widetilde{B} = 0$ to obtain a relation

$$\langle \widetilde{a}_i, \widetilde{Z}\rangle^2 = \sum_j \eta_j \langle a_j, Z\rangle^2 + \ell(Z) \tag{19}$$

for some coefficients $\eta_j$ and some affine function $\ell$ of $Z$. The following reasoning is now the same as in Kivva et al. [57], Sorrenson et al. [108]. We thus find that $\langle \widetilde{a}_i, \widetilde{Z}\rangle$ and its square can be written as polynimials of degree at most 2 in $Z$. This implies that in fact $\langle \widetilde{a}_i, \widetilde{Z}\rangle$ is an affine function of $Z$ (otherwise its square would be a quartic polynomial), i.e., we can write

$$\langle \widetilde{a}_i, \widetilde{Z}\rangle = \sum_j \lambda_j \langle a_j, Z\rangle + C_i = \langle \sum_j \lambda_j a_j, Z\rangle + C_i. \tag{20}$$

Equating the square of this relation with (19) and taking the gradient with respect to $Z$ (as a polynomial the function is differentiable) we find

$$2\sum_j \eta_j a_j \langle a_j, Z\rangle + w = 2\sum_j \lambda_j a_j \langle \sum_j \lambda_j a_j, Z\rangle + w' \tag{21}$$

for two vectors $w$ and $w'$. The equality (for $Z = 0$) implies $w = w'$. Now linear independence of $a_j$ implies that for each $r$

$$\eta_r a_r = \lambda_r \sum_j \lambda_j a_j. \tag{22}$$

Applying linear independence again we conclude that either $\lambda_r = 0$ or $\lambda_j = 0$ for all $j \neq r$. This implies that there is at most one $r$ such that $\lambda_r \neq 0$. The relation (20) and the bijectivity of $\varphi$ implies that there is exactly on $r(i)$ such that $\lambda_{r(i)} \neq 0$ and therefore

$$\langle \widetilde{a}_i, \widetilde{Z}\rangle = \lambda_{r(i)} \langle a_{r(i)}, Z\rangle + C_i. \tag{23}$$

Applying the same argument in the reverse direction we conclude that there is a permutation $\pi \in \mathfrak{S}_n$ such that

$$\langle \widetilde{a}_{\pi(i)}, \widetilde{Z}\rangle = \lambda_i \langle a_i, Z\rangle + C_i. \tag{24}$$

By linear independence we can find an invertible linear map $T$ such that

$$\widetilde{a}_{\pi(i)}^\top = a_i^\top T^{-1} \tag{25}$$

(i.e, $T^\top \widetilde{a}_{\pi(i)} = a_i$) and a vector $w \in \mathbb{R}^{d_z}$ (the $a_i$ are linearly independent) such that

$$\langle \widetilde{a}_{\pi(i)}, \widetilde{Z} \rangle = \lambda_i(\langle a_i, Z \rangle + \langle a_i, w \rangle). \tag{26}$$

In particular the relations (5) and (6) hold. Now it is straightforward to see that if $i \in S^e$, i.e., $a_i$ is a row of $A^e$ then $\widetilde{a}_{\pi(i)}$ is a row of $\widetilde{A}^e$ and vice versa. Indeed, this follows from (17) for environment $e$ together with (26) and linear independence of the atoms. Therefore we conclude from (25) that there is a permutation $P^e$ such that

$$\widetilde{A}^e = P^e A^e T^{-1}. \tag{27}$$

Moreover, (26) then implies setting $Z = f^{-1}(x)$, $\widetilde{Z} = \widetilde{f}^{-1}(x)$

$$\widetilde{A}^e \widetilde{f}^{-1}(x) = \Lambda^e P^e A^e (f^{-1}(x) + w) \tag{28}$$

holds for the same permutation matrix $P^e$ and a diagonal matrix $\Lambda^e$ whose diagonal entries can be related to (26). Let us assume now that row $k$ of $A^e$ is $a_i$ and row $k'$ of $\widetilde{A}^e$ is $\widetilde{a}_{\pi(i)}$. Now we consider the subspace $H \subset \mathbb{R}^{d_z}$ containing all $Z$ such that $\langle Z, a_j \rangle = 0$ for $j \neq i$. Via (26) this implies that $\langle \widetilde{a}_j, \widetilde{Z} \rangle$ is constant for $j \neq \pi(i)$. Then we conclude from (17) that for $Z \in H$

$$\frac{(\langle a_i, Z \rangle - b_k^e)^2}{2(\sigma_k^e)^2} = \frac{(\langle \widetilde{a}_{\pi(i)}, \widetilde{Z} \rangle - \widetilde{b}_{k'}^e)^2}{2(\widetilde{\sigma}_{k'}^e)^2} + c_k^e \tag{29}$$

for some constant $c_k^e$. Using (26) this implies that

$$\frac{(\langle a_i, Z \rangle - b_k^e)^2}{2(\sigma_k^e)^2} = \frac{(\lambda_i(\langle a_i, Z \rangle + \langle a_i, w \rangle) - \widetilde{b}_{k'}^e)^2}{2(\widetilde{\sigma}_{k'}^e)^2} + c_k^e. \tag{30}$$

Comparing the quadratic term and the linear term (note that $\langle a_i, Z \rangle$ can take any value on $H$) we find

$$\frac{1}{2(\sigma_k^e)^2} = \frac{\lambda_i^2}{2(\widetilde{\sigma}_{k'}^e)^2} \tag{31}$$

$$-\frac{b_k^e}{2(\sigma_k^e)^2} = -\frac{\lambda_i \widetilde{b}_{k'}^e - \lambda_i^2 \langle a_i, w \rangle}{2(\widetilde{\sigma}_{k'}^e)^2} \tag{32}$$

Combining the equation we obtain

$$\widetilde{b}_{k'}^e = \lambda_i(b_k^e - \langle a_i, w \rangle) \tag{33}$$

This implies then the relation

$$\widetilde{b} = \Lambda^e P^e (b + A^e w). \tag{34}$$

$\square$

## A.2  Proof of Theorem 1

In this section we prove our main Theorem 1. The proof is structured in several steps: First we remove the symmetries of the representation and derive the key relations underlying the proof. Then we show that we can identify the environment-concept matrix $M$ and then also the valuations collected in $B$. Once this is done we can complete the proof. We will need the following lemma to conclude the proof.

**Lemma 2.** *The relations* (3) *and* (6) *in Definition 4 define an equivalence relation of representations if we assume that the underlying atoms form a linearly independent set.*

The proof of this lemma can be found in Appendix A.3.

**Remark 3.** *Without the assumption on the underlying atoms the lemma is not true. In this case a slightly different scaling must be chosen (e.g., $(\Lambda^e)^{-1}\widetilde{b}^e = \Lambda^e P^e b^e - P^e A^e w$ instead of $\widetilde{b}^e = \Lambda^e P^e(b^e - A^e w)$). Since our results address the case of atoms we used the simpler definition in the main paper.*

We can allow slightly more general filtering distributions where $q$ is Gaussian with variance $\sigma_i^2$ if we filter on concept $i$, i.e., the variance needs to be constant for different environments and the same atom but might depend on the atom. The proof will cover this case, the simple case stated in the main paper is obtained by setting $\sigma_i^2 = \sigma^2$. Some steps of the proof (e.g., the expressions for the difference of the log-densities) agree with the proof of Theorem 2. To keep the proof self contained we repeat a few equations.

*Proof of Theorem 1.* We proceed in several steps.

**Step 1: Reduction to standard form.** Let us first transform every possible data representation into a standard form. Recall that we have the set of atomic concepts $\mathcal{C} = \{a_1, \ldots, a_n\}$. Recall that we defined the environment-concept matrix $M \in \mathbb{R}^{m \times n}$ in (7) and note that the natural generalisation reads

$$M_{ei} = \begin{cases} \frac{1}{\sigma_i^2} & \text{if } a_i \text{ is a row of } A^e, \\ 0 & \text{otherwise.} \end{cases} \tag{35}$$

We say that concept $a_n$ is conditioned on the environment $e$. Note that the nonzero entries of row $e$ of $M$ encode the set $S^e$. To pass from $A^e$ to its rows $a_i$ we assume that the $e$-th row of $A^e$ is $a_{i_j^e}$, i.e., $a_{i_j^e} = (A^e)^\top e_j$. Recall also consider the environment-valuation matrix $B$ which is given by

$$B_{ei} = \begin{cases} \frac{b_k^e}{\sigma_i^2} & \text{if } a_i \text{ is row } k \text{ of } A^e, \\ 0 & \text{otherwise.} \end{cases} \tag{36}$$

Denoting by $q_{\sigma^2}$ the centered Gaussian distribution with variance $\sigma^2$ we find in environment $e$

$$\ln(p(Z)) - \ln(p^e(Z)) = -\sum_{k=1}^{\dim(C_e)} \ln q_{(\sigma_k^e)^2}((A^e Z^e - b^e)_k) = \sum_{k=1}^{\dim(C_e)} \frac{(A^e Z^e - b^e)_k^2}{2(\sigma_k^e)^2} - c_e'$$

$$= \sum_{i=1}^n \frac{1}{2} M_{ei} \langle a_i, Z^e \rangle^2 - B_{ei} \langle a_i, Z^e \rangle - c_e. \tag{37}$$

Now we consider an invertible linear map $T : \mathbb{R}^{d_z} \to \mathbb{R}^{d_z}$ such that $T^{-\top} a_i = e_i$ for all $1 \le i \le n$. Such a map exists because we assume that the $a_i$ are linearly independent. Moreover, we consider a shift vector $\lambda \in \mathbb{R}^{d_z}$ with $\lambda_i = 0$ for $i > n$ which we fix later. We define $\Sigma \in \mathbb{R}^{d_z \times d_z}$ to be the diagonal matrix with entries $\Sigma_{ii} = \sigma_i$ for $1 \le i \le n$ and $\Sigma_{ii} = 1$ for $i > n$. Now we consider the linear map $L(z) = \Sigma^{-1} T z - \lambda$ and a new representation given by

$$\overline{z} = L(z), \quad \overline{f} = f \circ L^{-1}, \quad \overline{\mathcal{C}} = \{e_1, \ldots, e_n\}, \quad \overline{\sigma}_i = 1, \quad \overline{A}^e = A^e T^{-1},$$
$$\overline{p}(\widetilde{z}) = p(L^{-1}\widetilde{z})|\det T^{-1}|. \tag{38}$$

We also define

$$\overline{b}_k^e = \frac{b_k^e}{\sigma_i} - \lambda_i \quad \text{if row } k \text{ of } A^e \text{ is } a_i. \tag{39}$$

Define $\overline{M}$ and $\overline{B}$ in terms of $\overline{A}^e, \overline{b}^e$ and $\overline{\sigma}_i^2$ as before. We remark that all entries of $\overline{M}$ are either 0 or 1 and note that

$$\overline{M} = M\text{Diag}(\sigma_1^2, \ldots, \sigma_n^2) \tag{40}$$
$$\overline{B} = B\text{Diag}(\sigma_1^{-1}, \ldots, \sigma_n^{-1}) - M\text{Diag}(\lambda_1, \ldots, \lambda_n). \tag{41}$$

We claim that this model generates the same observations as the original model. By definition $L_* p = \overline{p}$ (as mentioned before, we slightly abuse notation and here refer to the distributions). Next, we calculate for any $\delta$

$$-2\ln q_1(\langle e_i, L(z) \rangle - \delta) = (\langle e_i, L(z) \rangle - \delta)^2$$
$$= (\langle e_i, \Sigma T z - \lambda \rangle - \delta)^2$$
$$= (\sigma_i^{-1} \langle T^\top e_i, z \rangle - \lambda_i - \delta)^2$$
$$= \frac{(\langle a_i, z \rangle - \sigma_i \lambda_i - \sigma_i \delta)^2}{\sigma_i^2}$$
$$= -2\ln q_{\sigma_i^2}(\langle a_i, z \rangle - \sigma_i \lambda_i - \sigma_i \delta). \tag{42}$$

Using this for $\delta = \overline{b}^e_k$ and some $k$ such that row $k$ of $A^e$ is $a_i$ we find

$$-2\ln q_1(\langle \boldsymbol{e}_i, L(z)\rangle - \overline{b}^e_k) = -2\ln q_{\sigma^2_i}(\langle a_i, z\rangle - \sigma_i\lambda_i - \sigma_i\overline{b}^e_k) = -2\ln q_{\sigma^2_i}(\langle a_i, z\rangle - b^e_k). \quad (43)$$

This then implies that for $\widetilde{z} = L(z)$

$$\prod_k q_1((\widetilde{A}^e\widetilde{z} - \widetilde{b}^e)_k) \propto \prod_k q_{\sigma^e_k}\left((A^e z - b^e)_k\right). \quad (44)$$

Combining this with the definition (2) and the definition $\overline{p}(\widetilde{z}) = p(L^{-1}\widetilde{z})|\det T^{-1}|$ we find that for $\overline{z} = L(z)$

$$\overline{p}^e(\widetilde{z}) \propto p^e(z) \quad (45)$$

and thus $\overline{f}(\overline{Z}^e) \overset{\mathcal{D}}{=} f(Z^e) \overset{\mathcal{D}}{=} X^e$. Moreover, one directly sees that the two representations are also equivalent in the sense of Definition 4. We now fix the vector $\lambda$ such that each row of $\overline{B}$ has mean zero. Finally, by changing the sign of $\widetilde{z}_i$ we can in addition assume that for every $i$ the first non-zero $\overline{B}_{ei}$ is positive. Finally we remark that Assumption 4 is still satisfied for $\overline{M}$ and $\overline{B}$. Indeed, $w^\top M = 0$ implies $w^\top \overline{M} = 0$ by (40). But then $w^\top \overline{B} = w^\top B\mathrm{Diag}(\sigma^{-1}_1, \ldots, \sigma^{-1}_n)$ by (41) which has all entries different from zero if this holds for $w^\top B$. In the following we will therefore always assume that the representation satisfies the properties of the $\overline{Z}$ variables and we remove the modifier in the following. The plan is now to show that $M$ and $B$ can be identified up to permutations of the rows (under the fixed normalization we derived in this step) and then show that every two representations with the same $M$ and $B$ can be identified.

**Step 2: The key identity** Let us here restate the key identity based on the difference of the log-densities. As is common in identifiability results for multi-environment data with general mixing we consider the difference in log densities. Consider

$$\begin{aligned}
\ln p^0(z) - \ln p^e(z) &= \sum_{i=1}^n \frac{1}{2}M_{ei}\langle \boldsymbol{e}_i, z\rangle^2 - B_{ei}\langle \boldsymbol{e}_i, z\rangle - c'_e \\
&= \sum_{i=1}^n \frac{1}{2}M_{ei}z_i^2 - B_{ei}z_i - c'_e
\end{aligned} \quad (46)$$

for some constant $c'_e$. Those functions will play a crucial role in the following and we will denote

$$g^e(z) = \ln p^0(z) - \ln p^e(z) \quad (47)$$

Note that since the log-density changes only by the Jacobian for pushforward measures we find that

$$g^e(z) = \ln p^0(z) - \ln p^e(z) = \ln p^0_X(f(z)) - \ln p^e_X(f(z)) = G^e(f(z)) = G^e(x). \quad (48)$$

Note that the functions $G^e(x)$ can be estimated from the distributions of $X^e$. We remark $X$ might be supported on a submanifold if $d_z$ and $d_x$ do not agree making the definition of the density subtle. But we can just consider any chart locally and consider the density of the pushforward with respect to the Lebesgue measure. The resulting difference expressed in $G^e$ will be independent of the chart as the determinant cancels thus $G^e$ is a well defined function. The relation

$$g^e(z) = G^e(f(z)) = G^e(x) \quad (49)$$

will be crucial in the following because it shows that properties of $g^e$ are closely linked to the identifiable functions $G^e$.

**Step 3: Identifiability of environment-concept matrix** Let us now show that we can identify which concepts are contained in which environment (up to relabeling of the concepts). Recall that $S^e = \{i \in [n] : a_i \text{ is a row of } A^e\}$ and we similarly define $S_T = \bigcup_{e\in T} S^e$ for all subsets $T \subset [m]$. The main observation is that we can identify $|S_T| = |\bigcup_{e\in T} S^e|$ for all subsets $T \subset [m]$. To show this we consider the set

$$I_T = \operatorname*{argmin}_z \sum_{e\in T} g^e(z). \quad (50)$$

Note that the function $g^e$ are convex functions, and they can be decomposed as sums of functions in $z_i$, i.e., for some functions $h_i^T$

$$\sum_{e \in T} g^e(z) = \sum_{i=1}^n h_i^T(z_i). \tag{51}$$

Now if $i \in S_T$ then $i \in S^e$ for some $e$ and thus $M_{ei} \neq 0$ for the $e$ and $h_i^T$ is the sum of quadratic function in $x_i$ which as a strictly convex function has a unique minimum $z_i^T$. On the other hand, if $i \notin S_T$ then $i \notin S^e$ for $e \in T$ and thus $M_{ei} = 0$ for all $e \in T$ and $h_i^T(z_i) = 0$. Thus we conclude that

$$I_T = \{z \in \mathbb{R}^{d_z} : z_i = z_i^T \text{ for } i \in S_T\}. \tag{52}$$

This is an affine subspace of dimension $d_z - |S_T|$. The relations $G^e(f(z)) = g^e(z)$ imply that

$$f(I_T) = \operatorname*{argmin}_x \sum_{e \in T} G^e(x). \tag{53}$$

Note that $G^e(x)$ is identifiable from the datasets $X^e$ and thus the submanifold (by assumption on $f$) $f(I_T)$ is identifiable and by finding its dimension we obtain $d_z - |S_T|$. Since $d_z$ is the dimension of the data manifold $f(X)$ we can indeed identify $|S_T|$ for all $T \subset [m]$. In particular, the total number of atomic concepts $n = |S_{[m]}|$ is identifiable (assuming that all atomic concepts are filtered upon at least once). Now, it is a standard result that we can identify the matrix $M$ up to permutation of the atomic concepts. Indeed, we can argue by induction in $m$ to show this. For $m = 1$ we just have $|S^1|$ atomic concepts appearing in environment 1 and $n - |S^1|$ concepts not appearing. For the induction step $m \to m+1$ we consider the sizes $|S_{T \cup \{m+1\}}|$ for $T \subset [m]$. Applying the induction hypothesis we can complete $M_{ei}$ for all columns such that $M_{m+1,i} = 1$. Similarly, we can consider the sizes $|S_T| - |S_{T \cup \{m+1\}}|$ to identify the matrix $M$ for concepts not used in environment $m+1$.

Thus, we can and will assume after permuting the atomic concepts that $M$ is some fixed matrix.

**Step 4: Identifiability of concept valuations** Next, we show that we can also identify the matrix $B$. We do this column by column, i.e., for one atomic concept after another. Assume we consider atomic concept $i$. Then we consider the set $T_i = \{e : M_{ei} = 0\}$ of concepts that not filter on atomic concept $i$. By Assumption 5 there is for every $i' \neq i$ an environment $e$ such that $i'$ is filtered on, i.e., $M_{ei'} \neq 0$. This implies $S_{T_i} = [n] \setminus \{i\}$. Then we consider as in (52) the set $I_{T_i}$ given by

$$I_{T_i} = \{z \in \mathbb{R}^{d_z} : z_{i'} = z_{i'}^{T_i} \text{ for } i' \in [n] \setminus \{i\}\}. \tag{54}$$

Note that all $z_{i'}$ for $i \neq i'$ are constant on $I_{T_i}$. Thus we find for any environment $e$ such that $i \in S^e$.

$$\begin{aligned}
g^e(z) &= \sum_{j=1}^n \frac{1}{2} M_{ej} z_j^2 - B_{ej} z_j - c'_e \\
&= \sum_{j \neq i}^n \frac{1}{2} M_{ej} z_j^2 - B_{ej} z_j - c'_e + \frac{1}{2} z_i^2 - B_{ei} z_i \\
&= c_{T_i, e} + \frac{1}{2} z_i^2 - B_{ei} z_i
\end{aligned} \tag{55}$$

on $I_{T_i}$ for some constant $c_{T_i}$.

Now we consider two concepts $e_1 \neq e_2$ such that atomic concept $i$ is contained in these two environments. Then we consider the set

$$I_{T_i}^{e_1} = \operatorname*{argmin}_{z \in I_{T_i}} g^{e_1}(z) = \{z \in \mathbb{R}^{d_z} : z_{i'} = z_{i'}^{T_i} \text{ for } i' \in [n] \setminus \{i\}, z_i = B_{e_1 i}\}. \tag{56}$$

Note that in the second equality we used that $g^{e_1}(z)$ depends on $z_i$ through $z_i^2/2 - B_{e_1 i} z_i$ so it is minimized at $B_{e_1 i}$. Now we find using (55)

$$\begin{aligned}
\min_{z \in I_{T_i}^{e_1}} g^{e_2}(z) - \min_{I_{T_i}} g^{e_2}(z) &= \min_{z \in I_{T_i}^{e_1}} c_{T_i, e_2} + \frac{1}{2} z_i^2 - B_{e_2 i} z_i - \min_{I_{T_i}} \left( c_{T_i, e_2} + \frac{1}{2} z_i^2 - B_{e_2 i} z_i \right) \\
&= c_{T_i, e_2} + \frac{1}{2} B_{e_1 i}^2 - B_{e_1 i} B_{e_2 i} - \left( c_{T_i, e_2} + \frac{1}{2} B_{e_2 i}^2 - B_{e_2 i}^2 \right) \\
&= \frac{(B_{e_1 i} - B_{e_2 i})^2}{2}.
\end{aligned} \tag{57}$$

As before, this quantity is identifiable from observations because $f(T_i)$ can be identified and we can minimize $G^{e_2}(x)$ over $f(T_i)$.

This allows us to identify $B_{e_1 i} - B_{e_2 i}$ up to a sign. However, we can evaluate this expression over all pairs $e_1$ and $e_2$ and pick the one with the maximal difference. Then all remaining values $B_{ei}$ for $e$ such that $i$ is filtered on in $e$ must satisfy $B_{ei} \in [B_{e_1 i}, B_{e_2 i}]$. Together with identifiability of $|B_{ei} - B_{e_1 i}|$ this allows us to identify all $B_{ei}$ up to one sign indeterminacy and a constant shift. However, in the first step we ensured that $\sum_e B_{ei} = 0$ for all $i$ which determines the shift and the sign is fixed by our choice of making the first non-zero entry positive. Thus, we can assume that our two representations have the same $M$ and $B$.

**Step 5: Identifiability of concepts**   We are now ready to prove our identifiability result.

Assume we have two representations $Z^e$, $f$, $p$ and $\widetilde{Z}^e$, $\widetilde{f}$, and $\widetilde{p}$ such that the corresponding environment-concept and environment-valuation matrices agree, i.e., $M = \widetilde{M}$ and $B = \widetilde{B}$. We consider the transition function $\varphi = \widetilde{f}^{-1} \circ f$ which is by assumption differentiable. What we want to show is that $\varphi(z)_i = z_i$ for all $z \in \mathbb{R}^{d_z}$ and $1 \leq i \leq n$. We now decompose $z = (z^c, z^o)$ into the concept part and the orthogonal part. We fix $z^o \in \mathbb{R}^{d_z - n}$ and define the function $\iota^o(z^c) = (z^c, z^o)$, the projection $\pi^c((z^c, z^o)) = z^c$, and $\varphi^o : \mathbb{R}^n \to \mathbb{R}^n$ given by $\varphi^o(z^c)_i = \varphi(\iota^o(z^c) = \varphi((z^c, z^o))_i$. Note that $\varphi^o$ is differentiable but not necessarily injective. Let us denote by $\boldsymbol{g} : \mathbb{R}^{d_z} \to \mathbb{R}^m$ the function with coordinates $\boldsymbol{g}_e = g^e$ and similarly we define $\boldsymbol{G} : M \to \mathbb{R}^d$. Identifiability will be based on the crucial relation

$$\boldsymbol{g}(\iota^o(z^c)) = \boldsymbol{G}(f(\iota^o(z^c))) = \boldsymbol{G}(\widetilde{f}(\varphi^o(z^c))) = \boldsymbol{g}(\varphi^o(z^c)). \tag{58}$$

Here we used in the last step that $g^e$ is defined in terms of $M$ and $B$ and thus agrees for both representations. Note that $\boldsymbol{g}$ is just a quadratic function. Differentiating we obtain

$$D_i g^e(z) = M_{ei} z_i - B_{ei}. \tag{59}$$

Concisely this can be written as

$$D\boldsymbol{g} = M \mathrm{Diag}(z_1, \ldots, z_n) - B. \tag{60}$$

Differentiating (58) we find

$$M \mathrm{Diag}(z_1, \ldots, z_n) - B = (M \mathrm{Diag}(\widetilde{z}_1, \ldots, \widetilde{z}_n) - B) D\varphi^o(z^c). \tag{61}$$

Let $v$ be a vector as in Assumption 4. Denote by $M^+ \in \mathbb{R}^{n \times m}$ the pseudoinverse of $M$ which has rank $n$ because $M$ has. We consider the matrix $\widetilde{M^+} \in \mathbb{R}^{n+1 \times m}$ given by

$$\widetilde{M^+} = \begin{pmatrix} M^+ \\ v^\top \end{pmatrix} \tag{62}$$

Let us multiply the relation (61) by $\widetilde{M^+}$ and find that

$$\begin{pmatrix} z_1 & & 0 \\ & \ddots & \\ 0 & & z_n \\ 0 & \cdots & 0 \end{pmatrix} - \widetilde{M^+} B = \left( \begin{pmatrix} \widetilde{z}_1 & & 0 \\ & \ddots & \\ 0 & & \widetilde{z}_n \\ 0 & \cdots & 0 \end{pmatrix} - \widetilde{M^+} B \right) D\varphi^o(z^c) \tag{63}$$

Note that the first $n$ rows of the left hand side are $\mathrm{Diag}(z_1, \ldots, z_n) - M^+ B$. This matrix is invertible for almost all values of $z^c = (z_1, \ldots, z_n)^\top$ because its determinant is a non-zero polynomial (the coefficient of the term $z_1 \cdot \ldots z_n$ is 1) which vanishes only on a set of measure zero. Outside of this set the left hand side of has rank $n$. Then the equality (63) implies that also the right hand side has rank $n$ and thus $D\varphi^o(z^c)$ has rank $n$ and thus is invertible. For $z^c$ outside of this set there is up to scaling a unique vector $w \neq 0$ (depending on $z_1, \ldots, z_n$ such that

$$w^\top \left( \begin{pmatrix} z_1 & & 0 \\ & \ddots & \\ 0 & & z_n \\ 0 & \cdots & 0 \end{pmatrix} - \widetilde{M^+} B \right) = 0 \tag{64}$$

From (63) we conclude using the invertibility of $D\varphi^o(z^c)$ that

$$w^\top \left( \begin{pmatrix} \widetilde{z}_1 & & 0 \\ & \ddots & \\ 0 & & \widetilde{z}_n \\ 0 & \dots & 0 \end{pmatrix} - \widetilde{M^+B} \right) = 0. \tag{65}$$

Next, we claim that for almost all values of $z^c$ the vector $w$ has all entries different from 0 (this property is invariant under rescaling). Actually we need this only for entries 1 to $n$ but the case $n+1$ is a bit simpler so we show it first. We show this by proving that for each entry $w_i$ there is only a null set of $z^c$ such that $w_i = 0$. Let $w = (w', 0)$ for some $w' \in \mathbb{R}^n$ and $w' \neq 0$, i.e., $w_{n+1} = 0$. Then

$$0 = w^\top \left( \begin{pmatrix} z_1 & & 0 \\ & \ddots & \\ 0 & & z_n \\ 0 & \dots & 0 \end{pmatrix} - \widetilde{M^+B} \right) = w'^\top \left( \mathrm{Diag}(z_1, \dots, z_n) - M^+B \right) \tag{66}$$

But this implies that $\mathrm{Diag}(z_1, \dots, z_n) - M^+B$ has non-trivial kernel, i.e., does not have full rank and we have seen above that this happens only for a subset of measure 0 of all $z^c$. Next we show that the same is true if $w_1 = 0$. Decompose $0 \neq w = (0, w')$. Then we find

$$0 = w^\top \left( \begin{pmatrix} z_1 & & 0 \\ & \ddots & \\ 0 & & z_n \\ 0 & \dots & 0 \end{pmatrix} - \widetilde{M^+B} \right) = w'^\top \left( \begin{pmatrix} 0 & z_2 & 0 & 0 \\ \dots & & \ddots & \\ 0 & & & z_n \\ 0 & \dots & \dots & 0 \end{pmatrix} - (\widetilde{M^+B})_{2:(n+1)} \right) \tag{67}$$

Thus we conclude that the matrix on the right hand side is not invertible. Its determinant is a polynomial in $z_2, \dots, z_n$ and its highest degree term is $\pm z_2 \cdot \ldots \cdot z_n \cdot (\widetilde{M^+B})_{(n+1),1}$. By definition of $\widetilde{M^+B}$ we find $(\widetilde{M^+B})_{(n+1),1} = (v^\top B)_1 \neq 0$ by Assumption 4 (recall that we showed invariance of the assumption under the transformation of $M$ and $B$). We find that the determinant is a non-zero polynomial and the set of its zeros is a set of measure 0 of all $z_2, \dots, z_n$ but since it does not depend on $z_1$ this holds true for almost all $z^c$. The same reasoning for $i = 2, \dots, n$ implies that for every $i$ the set of $z^c$ such that $w_i = 0$ is a set of measure zero. We have therefore shown that for almost all $z^c$ the rank of the left hand side of (63) is $n$ and the corresponding vector $w \neq 0$ has all entries different from zero. Subtracting (64) and (65) we obtain

$$0 = w^\top \begin{pmatrix} z_1 & & 0 \\ & \ddots & \\ 0 & & z_n \\ 0 & \dots & 0 \end{pmatrix} - w^\top \begin{pmatrix} \widetilde{z}_1 & & 0 \\ & \ddots & \\ 0 & & \widetilde{z}_n \\ 0 & \dots & 0 \end{pmatrix} = (w_1(z_1 - \widetilde{z}_1), \quad \dots \quad w_n(z_n - \widetilde{z}_n), 0). \tag{68}$$

Now $w_i \neq 0$ implies $z_i = \widetilde{z}_i$. We conclude that for almost all $z^c$ the relation $\varphi^o(z^c) = z^c$ holds. By continuity this implies that the relation actually holds everywhere. We conclude that $\pi^c \widetilde{f}^{-1} f((z^c, z^o)) = z^c$ for a fixed $z^o$ but since $z^o$ was arbitrary the relation holds for all $z^o$ and all $z^c$. Thus we conclude that for $1 \leq i \leq n$

$$\langle \boldsymbol{e}_i, \widetilde{f}^{-1}(x) \rangle = \langle \boldsymbol{e}_i, \varphi(f^{-1}(x)) \rangle = \langle \boldsymbol{e}_i, f^{-1}(x) \rangle \tag{69}$$

holds. This implies that those two representations satisfy (3) and (4) (with $P^e = \Lambda^e = \mathrm{Id}$ and $T = \mathrm{Id}$). But since this relation is an equivalence relation in our setting by Lemma 2 and since we showed equivalence to a representation in standard form in the first step we conclude that also any two representations are related through (3) and (4) thus finishing the proof. $\square$

## A.3 Remaining proofs

Here we prove the remaining auxiliary results.

*Proof of Lemma 1.* Since $M \in \mathbb{R}^{m \times n}$ has rank $n$ and $m = n+1$ there is exactly one vector $v \in \mathbb{R}^m$ such that $v^\top M = 0$ and $v \neq 0$. We claim that this vector has all entries different from zero.

Indeed suppose $v_m = 0$ which then implies $v_{1:(m-1)}^\top M_{1:(m-1)} = 0$. But by assumption every $n \times n$ submatrix of $M$ is invertible (this is equivalent to the rows being linearly independent) so we conclude that $v_{1:(m-1)} = 0$ which is a contradiction to $v \neq 0$. The same reasoning applies to every entry. Note that the assumption on $M$ implies that every column has at least one non-zero entry, i.e., every column of $B$ has one entry sampled from a continuous distribution. But then the probability that $v$ is orthogonal to a column is zero because this is a codimension 1 hyperplane of all valuations of this row (since all entries of $v$ are non-zero). $\square$

*Proof of Lemma 2.* Reflexivity is obvious, just pick $T = \mathrm{Id}$, $w = 0$, $\Lambda^e = P^e = \mathrm{Id}_{\dim(C^e)}$. To show symmetry we first consider the atoms. Let $\tilde{T} = T^{-1}$ and $\tilde{\pi} = \pi^{-1}$. Then

$$a_{\widetilde{\pi}(i)}^\top = a_{\pi^{-1}(i)}^\top T^{-1} T = \widetilde{a}_{\pi \circ \pi^{-1}(i)} \widetilde{T}^{-1} = \widetilde{a}_i \widetilde{T}^{-1}. \tag{70}$$

Let $\widetilde{w}$ be a vector such that for all $1 \leq i \leq n$

$$\langle a_i, w \rangle = -\frac{1}{\lambda_i} \langle \widetilde{a}_{\pi(i)}, \widetilde{w} \rangle. \tag{71}$$

Such a vector exists by linear independence of $\widetilde{a}_i$. Let $\widetilde{\lambda}_i = \lambda_{\widetilde{\pi}(i)}^{-1}$. Then we find that the relation (6), namely

$$\langle \widetilde{a}_{\pi(i)}, \widetilde{f}^{-1}(x) \rangle = \lambda_i \left( \langle a_i, f^{-1}(x) \rangle + \langle a_i, w \rangle \right) \tag{72}$$

implies

$$\langle a_{\widetilde{\pi}(i)}, f^{-1}(x) \rangle = \frac{1}{\lambda_{\widetilde{\pi}(i)}} \langle \widetilde{a}_{\pi \circ \widetilde{\pi}(i)}, \widetilde{f}^{-1}(x) \rangle - \langle a_{\widetilde{\pi}(i)}, w \rangle = \frac{1}{\lambda_{\widetilde{\pi}(i)}} \langle \widetilde{a}_i, \widetilde{f}^{-1}(x) \rangle + \frac{1}{\lambda_{\widetilde{\pi}(i)}} \langle \widetilde{a}_{\pi \circ \widetilde{\pi}(i)}, \widetilde{w} \rangle$$
$$= \widetilde{\lambda}_i (\langle \widetilde{a}_i, \widetilde{f}^{-1}(x) \rangle + \langle \widetilde{a}_i, \widetilde{w} \rangle). \tag{73}$$

It remains to be shown that this lifts to the concepts $C^e$. We first note that the relation (6) together with (71) and (3) implies that

$$\Lambda^e P^e A^e w = -\widetilde{A}^e \widetilde{w}. \tag{74}$$

Let $\widetilde{P}^e = (P^e)^{-1}$ and $\widetilde{\Lambda}^e = (P^e)^{-1} (\Lambda^e)^{-1} P^e$. Then (3) combined with the previous disply implies

$$A^e f^{-1}(x) = (P^e)^{-1} (\Lambda^e)^{-1} \widetilde{A}^e \widetilde{f}^{-1}(x) - A^e w$$
$$= \widetilde{\Lambda}^e \widetilde{P}^e \widetilde{A}^e \widetilde{f}^{-1}(x) + (P^e)^{-1} (\Lambda^e)^{-1} \widetilde{A} \widetilde{w} \tag{75}$$
$$= \widetilde{\Lambda}^e \widetilde{P}^e \widetilde{A}^e (\widetilde{f}^{-1}(x) + \widetilde{w}).$$

The relation

$$A^e = \widetilde{P}^e \widetilde{A}^e \widetilde{T}^{-1} \tag{76}$$

is a direct consequence of the definitions of $\widetilde{P}^e$ and $\widetilde{T}$ and (4) and the relation

$$b^e = \widetilde{\Lambda}^e \widetilde{P}^e (\widetilde{b}^e - \widetilde{A}^e w) \tag{77}$$

follows exactly as in (75). The proof of transitivity is similar (first establish the relations on the atomic concepts then lift it to $C^e$). $\square$

## B Comparison to Causal Representation Learning

In this appendix we describe causal representation learning and discuss the similarities and differences between the viewpoint taken in this paper and the standard setting in causal representation learning.

Causal Representation Learning (CRL) [102, 101] aims to learn representations of data that correspond to true causal generative processes. More precisely, if we assume that data $X$ is generated as $X = f(Z)$ where $Z$ are latent causal factors and $f$ is some arbitrary nonlinearity, the goal is to learn $f$ as well as the distribution of $Z$. Since the latent variables $Z$ are assumed to have causal relationships

among them, many works exploit the presence of interventional data to learn the generative model. CRL incorporates ideas from the field of causality [109, 86, 88, 95, 110] into the field of latent variable models and is a generalization of nonlinear independent component analysis [21, 45, 47] and disentangled representation learning [9, 88, 62]. The field has seen a surge of advances in the last few years, e.g., [53, 56, 33, 70, 60, 13, 78, 142, 36, 96, 123, 50, 49, 115, 124, 136, 133]. As motivated in Schölkopf et al. [102], CRL enables many desiderata such as robustness, out of distribution generalization, and in addition enables planning and alignment. CRL has also been successful in many domains such as computer vision [53, 126, 2], robotics [73, 11, 69, 140] and genomics [111, 139].

In our work, we take significant inspiration from this framework of causal representation learning and present a relaxed framework that is weaker, but more general and also importantly, aligns better with empirical works on interpretability of large pre-trained models in the literature. We now describe the setup of CRL more formally in Appendix B.1. Then, in Appendix B.2, we discuss conceptual differences between causal representation learning and our framework.

## B.1    Formal setup

We assume that we observe data $X \in \mathbb{R}^{d_x}$ with the generative model $X = f(Z)$ where $Z \in \mathbb{R}^{d_z}$ are the latent variables and $f$ is a deterministic mixing function. The dataset $X$ is sampled from a distribution $p$ and the goal is to recover the mixing function $f$ as well as the distributions of the underlying latent variables $Z_1, \ldots, Z_{d_z}$. To this end, this problem is over-parameterized since multiple pairs of $Z$ and $f$ could fit the dataset apriori, so the common practice in CRL is to impose various assumptions that will make this model *identifiable*. Here, identifiability is the notion that a unique set of parameters fit the model (up to trivial transformations). This makes the problem well-defined and feasible, although it could still be a hard problem to solve in practice. Below, we informally summarize two classes of prior works that enable such identifiability guarantees.

1. Disentangled representation learning: In this setting, we assume that the distributions of $Z_1, \ldots, Z_{d_z}$ are jointly independent. Different studies constrain the distribution of the variables $Z_1, \ldots, Z_{d_z}$, e.g., each $Z_i$ is independently sampled from $N(0, 1)$. This is also the setting studied in nonlinear independent component analysis [21, 45].

2. Causal Representation Learning: This setting is more general than the one above where we relax the independence assumption on the $Z_i$, and instead assume that they have (typically unknown) causal relationships among them. For instance, they could satisfy a linear structural causal model with Gaussian noise, i.e., $Z = AZ + \epsilon, \epsilon \sim N(0, I)$ where $A$ encodes a weighted directed acyclic graph. This setting is generalizes the previous setting, since having no causal relationships (i.e., $A = 0$) implies joint independence.

As explained earlier, in both these domains, a critical notion is that of identifiability [53, 25, 129], which posits that the given dataset(s) are diverse enough for the modeling assumptions, in order to ensure that a unique set of parameters fit the data. It's folklore that the disentangled representation learning model is not identifiable if all $Z_i$ are Gaussian [46, 71]. However, under appropriate assumptions, e.g., distributional, sparsity or observed side-information, the model becomes identifiable, see e.g., [53, 44, 11, 106, 60, 78, 141, 57, 13, 12, 142, 36, 96]. In addition, various works have proposed methods to learn them [33, 132, 26, 134, 67, 23, 13, 63, 14].

## B.2    Conceptual differences

In this section, we highlight the conceptual differences between causal representation learning and our framework.

**Are causal generative concepts necessarily interpretable?**    Moreover, we are constantly conjuring new concepts of interest since human-interpretable concepts are constantly evolving, e.g., the concept of mobile phones did not exist 100 years ago, but is a valid concept to learn now. Therefore, as opposed to working with a rigid model as in causal representation learning, we take the approach of working with a dynamic representation learning model. Finally, even if individual causal factors *are* interpretable (which may be the case in certain applications), the perspective that we take in this work is that the number of true generative factors could be prohibitively large so that attempting to extract

and interpret all of them together is infeasible, whereas the number of desired human-interpretable concepts is much smaller and more manageable.

**Number of environments needed**   When the ground truth generative process has ambient latent dimention $d_z$, for causal representation learning to be feasible, we usually require $d_z$ environments or datasets. For instance, in the iVAE setting [53] with $k$ sufficient statistics, we require $d_z k + 1 \geq d_z + 1$ environments. This is indeed necessary, as counterexamples show. However, it's not clear what the value of $d_z$ is for complex datasets, and it could potentially be prohibitively large.

But the question remains, do we need to learn the entire generative model for solving downstream tasks? Along these lines, there is a tremendous research effort attempting to relax such requirements by imposing various inductive or domain biases and by building a theory of partial identifiability [57, 69, 59]. This is for good reason, since even though it would be ideal to learn the full ground truth generative model, it may be prohibitively large and moreover it may not be necessary for the downstream tasks we care about, therefore it suffices to learn what is necessary. On this note, the related task of learning only a subset of the generative latent variables is also not easy as the latent variables interact in potentially complicated ways.

In this work, we show that if we only wish to learn $n \ll d_z$ concepts, it suffices to have $O(n)$ environments instead of $\Omega(d_z)$ environments. Therefore, our results can be viewed as a result on partial identifiability with a sublinear number of environments.

**Multi-node interventions**   Multi-node interventions are an exciting area of study in CRL, since they are a natural extension of existing works and are more useful for modeling various real-life datasets where it can be hard to control precisely one factor of variation. This is easily incorporated in our setting by utilizing non-atomic concepts, since each non-atomic concept is a collection of vectors corresponding to atomic concepts and can be modified simultaneously by changing the valuation.

**Conditional vs. interventional data**   In this work we focus on conditional data and identification of concept structure, while a recent trend in CRL is to focus on interventional data and identification of the causal structure [110, 122, 14, 50, 126]. For causal models, interventions are a natural approach to solving the identifiability problem, however, in the absence of an assumed causal model (as in our framework), interventions may not even be formally well-defined. In our framework, we do not think of concepts as being causal variables that are connected by a graph. (We note that an interesting approach would be to study learning concepts over a given causal generative model, which is an intriguing direction for future study that we do not pursue in this work).

By contrast, conditional data does not require the formal framework of causal models, and is often more frequently available in practice. Conditional data can be obtained by selection through filtering, e.g., patients that are admitted to different hospitals based on the severity of their condition or by the availability of label information as in the CLIP setting [92]. Thus conditional data can be obtained by observing the system in different condtions. On the other hand interventional data requires manipulation of the system which is more difficult to obtain in general.

## C   Alternate definitions of concept conditional measure

In this section, we present alternate feasible definitions for data distributions than the one we introduced in Section 4.2. While we went with the definition most suited for practice, these alternate definitions are also justifiable in different scenarios and are exciting avenues for further study.

We want to essentially define a concept $C$ via a conditional measure $p_C$ where the concept $C$ is identified with an affine subspace $C = \{Z \in \mathbb{R}^{d_z} : A^C Z = b^C\}$ for some $A^C \in \mathbb{R}^{k \times d_z}$, $b^C \in \mathbb{R}^k$. We consider the shifted parallel linear subspace $C_0 = \{Z : A^C Z = 0\}$ and the orthogonal splitting $\mathbb{R}^{d_z} = C_0 \oplus V$. Suppose we have a distribution $q_V$ on the space $V$ which will typically be a Gaussian centered around $v^C \in V$ which is the unique solution of $A^C v^C = b^C$. In addition we have a base distribution $p$ on $\mathbb{R}^{d_z}$. We will assume that all distributions have a smooth density so that conditional probabilities are pointwise well defined. There are at least three ways to create the context conditional measure $p_C$.

1. The first option is to enforce that the distribution of the $V$ marginal $p_C(v) = \int_{C_0} p_C(v, c)\, dc$ exactly matches $q_V(v)$ while the in-plane distribution $p_C(c|v = v_0) \propto p_C(c, v_0)$ remains invariant, i.e., equals $p(c|v = v_0)$. Under this condition, there is a unique measure $p_C$ given by

$$p_C(c, v) \propto q_V(v) \frac{p(c, v)}{\int_{C_0} p(c', v)\, dc'}.$$

   In other words, to get $(c, v)$ we sample $v \sim q_V$ and then $c \sim p(c|v)$ according to the conditional distribution.

2. The second option is to again enforce the $V$ marginal but instead of keeping the in plane distribution we average over the $V$ space. Then we obtain

$$p_C(c, v) \propto q_V(v) \int_V p(c, v')\, dv'.$$

   This corresponds (vaguely) to a $\mathrm{do}(v)$ operation from causal inference, i.e., we sample according to $p(v, c)$ and then do a random intervention on $v$ with target distribution $q_V$.

3. The third option is to take a Bayesian standpoint. Then we view $p$ as a prior and $q_V$ as the context dependent acceptance probability, i.e., we sample by $p$ and then accept with probability $q_V$. Then we find

$$p_C(c, v) = \frac{p(c, v) q_V(v)}{\int p(c, v) q_V(v)\, dv\, dc} \propto p(c, v) q_V(v). \tag{78}$$

   This is probably the closest aligned to practice, so this is the one we study in this work. To justify this option, imagine the following scenario. If we wish to learn the concept of *red color*, a first step would be to curate a dataset of red objects. To do this, we first consider a collection of photos of objects of varying color and then filter out the ones that look red. The concept conditional measure we define aligns with this process. To learn the actual red concept accurately, our theory predicts that it is sufficient to have additional datasets of objects that are not red, from which we can distinguish red objects, thereby learning the concept of red color.

The next question is how to define the measure $q_V$. When considering a single concept $A^C Z = b^C$ the most natural option to consider $N(v^C, \sigma^2 \mathrm{Id}_V)$ where $v^C \in V$ is the unique solution of $A^C v^C = b^C$ and $\sigma > 0$ is a positive constant. This is what we do in this work (note that $\sigma^2$ can be set to 1 by scaling the concept and valuation accordingly).

However, we can also use alternate definitions as suggested above. For instance, we can set $AZ \overset{\mathcal{D}}{=} N(b^C, \mathrm{Id})$. Then $Z \sim N(v^C, (A^\top A)^{-1})$. However, this runs into some technical issues we sketch (and leave to future work to handle this). Consider the intersection of multiple concepts $C^e$. In this case the concept space is given by the intersection $C = \bigcap C^e$ and $C_0 = \bigcap (C^e)_0$ and we have the orthogonal decomposition $\mathbb{R}^{d_z} = C_0 \oplus \sum V^e$. In general the spaces $V^e$ are not necessarily orthogonal but it is reasonable to assume that the non-degeneracy condition $\dim(\sum V^i) = \sum \dim(V^e)$ holds. Now set $V = \sum V^e$. If we choose just the standard normal distribution for $q_{V^e}$ we can define just as in our approach

$$q_V \sim N(v^C, \sigma^2 \mathrm{Id}_V). \tag{79}$$

The second option is to enforce that the marginals of $q_V$ agree with $q_{V^e}$, i.e., $q_V(\Pi_{V^e}(v) \in O) = q_{V^e}(O)$ for $O \subset V^e$. This results in the set of equations for all $i$

$$A^e \Sigma (A^e)^\top = \mathrm{Id}_{V^e}. \tag{80}$$

It is likely that this system has a unique solution when non-degeneracy holds for $V^e$ and this is clearly true for orthogonal spaces but it is not clear how to solve this in general.

## D   Analysis of pretrained CLIP models

In this section we provide additional experimental details and further results for the analysis of pretrained CLIP models [92].

### D.1 Experimental Details

We transform the images from the 3d-Shapes dataset to match the CLIP training data, i.e., reshape to images of size 224 and match the channel distributions. Then we calculate the embeddings for all images in the dataset using two CLIP models, a model with a vision transformer backbone ('ViT-B/32') and a model with a Resnet backbone ('RN101')[3]. We split the embedded images in to training and test sets of equal size. Then for any factor of variation (orientation of the scene, shape and scale of the object, and hue of floor, wall, and object) we perform the following procedure. For each pair of values of a factor of variation we run logistic regression on the embeddings for those two values of the concept to classify which value is taken for a given embedding. We average the directions of the logistic regression vectors $\beta_i$, i.e., consider $\bar{\beta} = N^{-1} \sum_{i=1}^{N} \beta_i$. Since the direction is defined only up to a sign (depending on the order of the two groups) we repeatedly replace $\beta_i$ by $-\beta_i$ if the scalar product with the current mean is negative (this is a heuristic procedure to align $\beta_i$ with $\bar{\beta}$). We then use the learned concept vectors $a = \bar{\beta}$ to evaluate the concept valuations on the held out test data, i.e., we evaluate $\langle a, Z \rangle$ where $Z = f^{-1}(X)$ is the embedding of an image $X$. The preprocessing to calculate the CLIP image embeddings required few hours on a A100-GPU. The remaining evaluations were performed on a standard notebook.

### D.2 Further results

Here we report the mean and standard deviations of the per-class concept valuations $\langle a, Z \rangle$ for the concept vectors learned as described in Section D.1. The results for the six factors of variation can be found in Tables 2, 3, and 4. We observe that shape, scale, and orientation are well aligned with linear subspaces. For the hue variables this still holds to some degree the discrepancy might be attributed to hue not being an atomic concept (colours are typically represented by at least two numbers). Moreover, we consider the correlation coefficient of the valuastions obtained for different embedding models, i.e., for $\langle a^{M_1}, Z_i^{M_1} \rangle$ and $\langle a^{M_2}, Z_i^{M_2} \rangle$ where $a^{M_1}$ and $a^{M_2}$ are concept vectors for the same concept and two different models and $Z_i^{M_1}$ and $Z_i^{M_2}$ denote the embeddings of the two models $M_1$ and $M_2$ of sample $X_i$. We report these correlation coefficients for the two CLIP models in Table 5. The results indicate that the valuations indeed approximately agree up to a linear transformation. Note that for the scene orientation attribute the valuation corresponds to the absolute value of the angle.

Table 2: Mean valuations and standard deviation on the test set for the floor hue and wall hue attributes.

| Floor hue | Vit-B/32 | RN101 | Wall hue | Vit-B/32 | RN101 |
|---|---|---|---|---|---|
| 0.0 | $-1.4 \pm 1.4$ | $-0.3 \pm 0.9$ | 0.0 | $1.1 \pm 1.3$ | $-1.5 \pm 1.4$ |
| 0.1 | $4.5 \pm 1.5$ | $1.4 \pm 0.8$ | 0.1 | $2.8 \pm 1.3$ | $1.8 \pm 1.0$ |
| 0.2 | $4.3 \pm 1.3$ | $3.2 \pm 0.8$ | 0.2 | $3.3 \pm 1.1$ | $1.5 \pm 0.9$ |
| 0.3 | $2.2 \pm 1.4$ | $3.0 \pm 0.8$ | 0.3 | $1.7 \pm 1.0$ | $0.8 \pm 0.8$ |
| 0.4 | $1.2 \pm 1.5$ | $2.2 \pm 0.8$ | 0.4 | $0.8 \pm 1.3$ | $0.5 \pm 0.9$ |
| 0.5 | $0.0 \pm 1.1$ | $0.5 \pm 0.8$ | 0.5 | $-0.6 \pm 1.2$ | $-0.6 \pm 1.1$ |
| 0.6 | $-2.8 \pm 1.3$ | $-0.4 \pm 0.9$ | 0.6 | $-3.3 \pm 1.2$ | $-2.3 \pm 1.1$ |
| 0.7 | $-5.8 \pm 1.5$ | $-2.0 \pm 1.0$ | 0.7 | $-3.6 \pm 1.2$ | $-3.7 \pm 1.0$ |
| 0.8 | $-3.8 \pm 1.4$ | $-1.3 \pm 0.9$ | 0.8 | $-1.4 \pm 1.1$ | $-2.0 \pm 1.0$ |
| 0.9 | $-3.2 \pm 1.4$ | $-1.0 \pm 0.8$ | 0.9 | $-0.6 \pm 1.2$ | $-2.0 \pm 1.1$ |

## E  Inference-Time Intervention of Large Language Models

In this section, we first briefly describe Large Language Models and the recent Inference-Time Intervention (ITI) technique proposed for LLM alignment, which we build on. Then, we use our framework to provide better intuition on some intriguing observations about ITI, including why it works. And then we exploit our ideas to improve the performance of ITI by choosing the steering direction to be a matrix instead of a vector.

---

[3]Models are publicly available under `https://github.com/openai/CLIP`

Table 3: Mean valuations and standard deviation on the test set for the object hue and scene orientation attributes.

| Scene orientation (°) | Vit-B/32 | RN101 |
|---|---|---|
| -30.0 | $-4.9 \pm 1.4$ | $-0.0 \pm 1.1$ |
| -25.7 | $-4.0 \pm 1.3$ | $0.4 \pm 1.2$ |
| -21.4 | $-2.9 \pm 1.3$ | $-0.8 \pm 1.2$ |
| -17.1 | $-0.2 \pm 1.4$ | $-1.4 \pm 1.1$ |
| -12.9 | $3.3 \pm 1.5$ | $-3.9 \pm 1.1$ |
| -8.6 | $7.5 \pm 2.1$ | $-6.7 \pm 0.9$ |
| -4.3 | $7.2 \pm 2.4$ | $-7.4 \pm 1.1$ |
| 0.0 | $8.2 \pm 2.7$ | $-8.2 \pm 1.2$ |
| 4.3 | $5.8 \pm 2.3$ | $-7.6 \pm 1.1$ |
| 8.6 | $6.5 \pm 1.9$ | $-7.0 \pm 1.0$ |
| 12.9 | $2.0 \pm 1.6$ | $-4.7 \pm 0.9$ |
| 17.1 | $-2.9 \pm 1.3$ | $-2.2 \pm 0.9$ |
| 21.4 | $-4.8 \pm 1.3$ | $-1.8 \pm 1.1$ |
| 25.7 | $-5.7 \pm 1.5$ | $-0.7 \pm 1.1$ |
| 30.0 | $-6.6 \pm 1.8$ | $-0.7 \pm 1.1$ |

| Object hue | Vit-B/32 | RN101 |
|---|---|---|
| 0.0 | $-0.3 \pm 1.5$ | $-0.1 \pm 1.1$ |
| 0.1 | $4.8 \pm 2.1$ | $1.4 \pm 1.0$ |
| 0.2 | $6.0 \pm 2.0$ | $2.7 \pm 0.8$ |
| 0.3 | $3.9 \pm 1.7$ | $2.6 \pm 0.7$ |
| 0.4 | $2.3 \pm 1.4$ | $2.2 \pm 0.7$ |
| 0.5 | $-0.5 \pm 1.6$ | $0.3 \pm 0.9$ |
| 0.6 | $-4.8 \pm 1.8$ | $-1.8 \pm 0.9$ |
| 0.7 | $-5.6 \pm 1.9$ | $-2.4 \pm 1.0$ |
| 0.8 | $-3.4 \pm 1.4$ | $-1.3 \pm 0.9$ |
| 0.9 | $-1.9 \pm 1.4$ | $-0.6 \pm 1.0$ |

Table 4: Mean valuations and standard deviation on the test set for the scale and shape attributes.

| Scale | Vit-B/32 | RN101 |
|---|---|---|
| 0.8 | $10.6 \pm 2.6$ | $7.0 \pm 1.5$ |
| 0.8 | $8.3 \pm 2.1$ | $5.2 \pm 1.4$ |
| 0.9 | $5.0 \pm 1.9$ | $3.6 \pm 1.3$ |
| 1.0 | $1.9 \pm 1.9$ | $1.8 \pm 1.1$ |
| 1.0 | $-1.3 \pm 1.8$ | $0.2 \pm 1.1$ |
| 1.1 | $-4.3 \pm 2.0$ | $-1.4 \pm 1.2$ |
| 1.2 | $-7.1 \pm 2.1$ | $-2.8 \pm 1.2$ |
| 1.2 | $-9.3 \pm 2.3$ | $-3.9 \pm 1.3$ |

| Shape | Vit-B/32 | RN101 |
|---|---|---|
| Cube | $8.2 \pm 1.4$ | $6.9 \pm 0.9$ |
| Cylinder | $2.9 \pm 1.6$ | $2.9 \pm 0.9$ |
| Ball | $-3.6 \pm 1.6$ | $-1.2 \pm 0.7$ |
| Ellipsiod | $-11.8 \pm 3.1$ | $-5.5 \pm 1.7$ |

### E.1 Preliminaries

**Large Language Models (LLMs)**  LLMs are large models capable of generating meaningful text given a context sentence. Due to large-scale training, modern LLMs have shown remarkable capabilities and achieve expert-human-like performance in many benchmarks simultaneously. The architecture of many generative pre-trained transformers (GPT)-style LLMs consists of several transformer layers stacked on top of each other. Since we'll be intervening on them during inference, we'll describe the transformer architecture [125, 29] briefly here. First, the sequence of input tokens (tokens are sub-word units) are encoded into a vector $x_0$ using a (learned) text embedding matrix and in many cases also a positional embedding matrix. Then, a series of transformer layers act on this vector which passes through a residual stream, to obtain vectors $x_0, x_1, \ldots, x_n$. The final vector $x_n$ is then decoded back into token probabilities with a (learned) unembedding matrix. Each transformer layer consists of a multi-head attention mechanism and a standard multilayer perceptron, which captures the nonlinearity.

In the $l$th layer, each single multi-head attention mechanism can be described as

$$x_{l+1} = x_l + \sum_{h=1}^{H} Q_l^h x_l^h, \qquad x_l^h = \text{Att}_l^h(P_l^h x_l)$$

Here, $P_l^h$ and $Q_l^h$ are matrices that linearly map the vector to an activation space and back respectively, and Att denotes the attention mechanism that allows communication across tokens. Here, we have kept the notation consistent with Li et al. [66] for the sake of clarity.

Table 5: Correlation coefficients of the evaluations learned for two different CLIP models evaluated on the full dataset.

| Concept | $\rho$ |
|---|---|
| Floor hue | 0.86 |
| Wall hue | 0.83 |
| Object hue | 0.86 |
| Scale | 0.53 |
| Shape | 0.95 |
| Orientation | -0.70 |

In our setting, we consider the entire set of activations as the learnt latent vector $Z$. That is, the input is $x = x_0$ and the pre-trained model is essentially the function $f$ such that $f(x)$ consists of the concatenation of the vectors $\{x_l\}_{l \geq 1}$, the intermediate activations $\{x_l^h\}_{l \geq 0}$ and also the output of the linear transformations $\{P_l^h x_l\}_{l \geq 0}, \{Q_l^h x_l^h\}_{l \geq 0}$. Our theory hinges on the assumption that pre-trained LLMs satisfy the linear representation hypothesis, that is, various relevant concepts can be realized via linear transformations of the latent transformation $f(x)$. Indeed, this has been empirically observed to hold in many prior works [17, 118, 81, 79, 66, 85, 38, 52] (see also related works on geometry of representations [51, 52] and references therein). It's a fascinating question why such models trained with next token prediction loss also learn linear representations of various human-interpretable concepts such as sentiment, see Jiang et al. [52] for recent progress on this problem.

It's well-known that despite large-scale pretraining and subsequent improvement of pre-trained models via techniques like Reinforcement Learning with Human Feedback (RLHF) and Supervised Fine-Tuning (SFT) [84, 6, 119], significant issues still remain [107], e.g., the model can hallucinate or generate incorrect responses (even though the model *knows* the correct response which can be extracted via other means, e.g., Chain-of-Thought prompting [131]). Various methods have been proposed to fine-tune the models [84, 6, 7, 119, 93] but many of them are expensive and time- and resource-intensive as they requires huge annotation and computation resources. Therefore, more efficient techniques are highly desired, one of which is the category of methods known as activation patching. activation patching (also called activation editing or activation engineering) [40, 128, 112, 121, 143, 138, 65, 76].

**Inference-Time Intervention, an activation patching method for truthfulness**    Activation patching is a simple minimally invasive technique to align LLMs to human-preferences. Specifically, given various concepts such as truthfulness, activation patching makes modifications to the model during inference time so that the desired concepts can be aligned. This technique can be thought of as an application of the emerging field of mechanistic interpretability [83], which aims to interpret the learnt latent vector in terms of human-interpretable concepts, thereby allowing us to reverse-engineer what large models learn.

Activation patching has many variants [65, 40, 76], but we'll focus on the simple technique of adding *steering vectors* to various intermediate layers during intervention [112, 121, 66, 98]. This means that during inference, the output activations are modified by adding a constant vector in order to promote alignment of some concept. The vector will be learnt independently based on separate training data.

In particular, a recent technique called Inference-Time Intervention (ITI) was proposed to do this for the specific concept of truthfulness. ITI focuses on the activation heads $\{\text{Att}_l^h(P_l^h x_l)\}_{l \geq 0}$ and add to them steering vectors in order to promote truthfulness. To learn the steering vectors, a subset of the TruthfulQA dataset [68], namely a dataset of questions $q_i$ with annotated true $(a_{i,j}, 0)$ and false answers $(a_{i,j}, 1)$, are prepared as $\{q_i, a_i, y_i\}_{i=1,2,...}$. For each sample, the question and answer are concatenated as a pair and the corresponding activations of the heads $x_l^h$ (for the final token) are computed via forward passes. Then, a linear probe $\text{sigmoid}(\langle \theta, x_l^h \rangle)$ is independently trained on each activation head to distinguish true from false answers. Finally, the top $K$ heads based on the accuracy of this classification task are chosen (for a tunable hyperparameter $K$) and the steering vector $\theta_l^h$ for

the $h$-th head in layer $l$ is chosen to be the mean difference of the activations between the true and false inputs. The intuition is that this direction roughly captures the direction towards truthfulness.

Formally, for the $h$th head of the $l$th layer, ITI adds the steering vector $\alpha \sigma_l^h \theta_l^h$ so as to get

$$x_{l+1} = x_l + \sum_{h=1}^{H} Q_l^h(x_l^h + \alpha \sigma_l^h \theta_l^h), \qquad x_l^h = \text{Att}_l^h(P_l^h x_l)$$

during inference. Here, $\theta_l^h$ is the steering vector, $\sigma_l^h$ is the standard deviation of the activations of this head along the chosen direction and $\alpha$ is a hyperparameter. That is, the activations are shifted along the truthful directions by a multiple of the standard deviation, and this is repeated autoregressively. Note that this does not depend on the specific GPT-like model being used. The intuition is that during inference, the activations are intervened upon to shift towards the truthful direction. The top $K$ heads are chosen to be minimally intrusive and also a design choice based on observations of the probing metrics.

**Performance of ITI**  In Li et al. [66], ITI was shown to significantly improve the truthfulness of various LLMs after having been trained on as few as a few dozen samples, compared to what's needed for Reinforcement Learning based techqniues [84, 34]. ITI was evaluated on the TruthfulQA benchmark [68], which is a hard adversarial benchmark to evaluate truthfulness of language models. In particular, it contains 817 questions with a multiple-choice and generation tracks, spanning 38 categories such as logical falsehoods, conspiracies and common points of confusion. For the multiple-choice questions, the accuracy is determined by the conditional probabilities of candidate answers given the question. Evaluating the generation track questions is harder, and it is done by generating a model output and then evaluating it via a finetuned GPT-3-13B model [68, 80]. Moreover, the choice of the intervention strength $\alpha$ is calibrated so that it's neither too small (to promote truthfulness) nor too large (to ensure the original capabilities of the LLM are not lost). To check if the original capabilies are preserved, [66] compute two additional quantities to measure how far the modified model deviates from the original model. These are the Cross-Entropy (CE) loss, which is standard in language modeling and the Kullback–Leibler divergence (KL div.) of the next token probabilities before and after intervention. To compute these quantities, a subset of Open Web Text is used [91]. Finally, it was shown that ITI implemented on the LLaMA [119], Alpaca [116] and Vicuna [20] models significantly improved their performance on the TruthfulQA benchmark compared to the baseline models. Moreover, in many cases, it also beat other techniques such as few-shot prompting and supervised fine-tuning. Please see Li et al. [66] for additional details.

### E.2 Interesting observations of ITI

While the elegant ITI technique was designed to align LLMs towards truthfulness in practice, it also raised fascinating and intriguing questions in mechanistic interpretability. In addition to improving the technique of ITI itself, our work makes progress towards some of these questions via our framework.

1. The authors of Li et al. [66] state in section 2 that although the technique works well in practice, it's not clear what ITI does to the model's internal representations. In addition, prior works [17, 118, 81, 79, 85, 52] have observed empirically that the latent representations learned by LLMs seem to have interpretable linear directions, which ITI exploits. We use our framework to illustrate in more detail one possible explanation of what ITI does to the model representations and why it works, in the next section.

2. The authors visualize the geometry of "truth" representations in section 3.2 of their work via the following experiment: For the most significant head (layer 14, head 18), after finding the first truthful direction via the linear probing technique, they remove it and attempt to find a second probe orthogonal to the first. They find surprisingly that the second probe is also very informative, leading them to predict that the concept of "truth" lies in a subspace, not a single direction. Restated in our framework, the concept of truthfulness is a non-atomic concept (as per Definition 2). This served as an inspiration for our proposed technique in the next section, where we propose to use steering matrices instead of steering vectors for LLM alignment.

3. As $\alpha$ was increased, the authors observed that truthfulness of the model increased however helpfulness decreased. This suggests that the "truthfulness" and "helpfulness" concepts

are not atomic (as per Definition 2) however they share certain atomic concepts. We leave to future work the exciting question of mechanistically extracting such common atomic concepts.

### E.3    The choice of the steering vector

In this section, we will use our theoretical framework to get insights about the ITI technique and use it to improve alignment. First, similar to the multimodal CLIP setting, we will assume that the non-linearity has already been learned up to a linear transformation (by large-scale training of LLMs). This aligns with our theoretical insights because the training data for powerful LLMs are diverse, so they essentially satisfy our core assumptions (see also the related work [37] that proposes that context is environment in LLM training). Therefore, we simply focus on the downstream tasks, which in this section is LLM alignment. The difficulty, of course, is that we do not know the concept matrix nor the valuations.

We will now analyze the truthfulness concept via our framework and give more insight on why the mean of the differences is a reasonable choice of steering vector for ITI. Based on our theory, we will then provide a modification to this choice that uses steering matrices instead of steering vectors. Since this section is based on heuristics and informal assumptions, we will refrain from making any formal claims or analyses. Indeed, a formal analysis of concepts in natural language is a hard problem in general and we do not attempt it here. We conclude with ideas for potential extensions that're worth exploring in future work.

Denote the function $h$ to be the sequence of head activations $h(x) = (x_l^h)_{l,h} \in \mathbb{R}^d$. Note that while we can study general steering vectors for the entire latent space of representations $f(x)$ learned by LLMs as some works do, ITI focuses only on steering the head activations $h(x)$, so we will apply our framework to this subset representation space. In addition, we will make the simplification that we neglect the effects of the steering vector from bottom layers towards the top layers, which we do because we are dealing with sparse steering vectors and also, each single head shift is minor and does not in isolation change the behavior of the model as verified by experiments [66][Appendix B.1].

Applying our framework, we model the concept of truth via the concept matrix $A \in \mathbb{R}^{d_C \times d}$ and two valuations $b_0, b_1 \in \mathbb{R}^{d_C}$ corresponding to *False* and *True* respectively. In other words, the set of false sentences and true sentences lie respectively in

$$\mathcal{S}_{false} = \{x | Ah(x) = b_0\}, \qquad \mathcal{S}_{true} = \{x | Ah(x) = b_1\}$$

Note that they only approximately lie in these spaces because of our notion of concept conditional distribution. However, if we reasonably assume that the Gaussian concentration region is much smaller than the separation between these hyperplanes, then the rest of the arguments in this section should apply.

Now, a steering vector $\eta$ is a vector such that it moves the activations from the false space to the true space, while keeping other concepts unaffected. That is, if we pick a false sentence $x$, i.e., $Ah(x) = b_0$, then the steering vector $\eta \in \mathbb{R}^d$ essentially steers the activations so that $A(h(x) + \eta) = b_1$. In other words, it moves the sentence from false to true. Indeed, many vectors $\eta$ do satisfy this equality, because we could move $h(x)$ to any point in the hyperplane $\{AZ = b_1\}$. Therefore the goal is to find an optimal $\eta$ that does not (significantly) affect other concepts of interest, i.e., $B(h(x) + \eta) \approx Bh(x)$ (equivalently $B\eta = 0$) for any other concept of interest $B$. Indeed, a natural choice of the steering vector will be $A^+(b_1 - b_0)$ where $A^+$ is the pseudoinverse of $A$. This vector will precisely affect this concept space and will not affect the concept valuations for any concept orthogonal to $A$. However, there are two issues with this approach: We do not know $A$ and therefore we will approximate this steering vector from training samples and there is no guarantee that other concepts of interest are orthogonal to $A$ (note that angles between concepts are not even identifiable).

Previous approaches are based on a collection of counterfactual sentence pairs $c_i^F, c_i^T$ which correspond to a false answer and a true answer for the same question $q_i$. Consider the $i$th counterfactual pair $c_i^F, c_i^T$. We will assume the reasonable scenario that the only difference among their concepts is the concept of truthfulness. That is, for any other concept of interest $B_i$ for this sample the valuations of $B_i$ for these pairs $c_i^F$ and $c_i^T$ are identical. A common strategy is to use the mean

$$\eta = \frac{1}{n} \sum_{i=1}^{n} h(c_i^T) - h(c_i^F) \tag{81}$$

as a steering vector. Note that if

$$A(h(c_i^T) - h(c_i^F)) \approx b_1 - b_0, \tag{82}$$

i.e., the truthfulness valuation is changed as desired for all samples then

$$A\eta = b_1 - b_0. \tag{83}$$

Moreover, concepts of interest are preserved in two prototypical settings. First, if concepts of interest are the same for all samples and the new datapoint, i.e., $B = B_i = B_j$ in which case

$$B\eta = \frac{1}{n} \sum_{i=1}^{n} B_i(h(c_i^T) - h(c_i^F)) = 0. \tag{84}$$

Similarly, if concepts of interest for a new point $x$ are $B_x$ and the valuations of $B_x(h(c_i^T) - h(c_i^F))$ of the counterfactual pairs are random, independent, and centered, then we expect them to approximately cancel and

$$B_x\eta \approx 0. \tag{85}$$

Note that in this case, this is not true if just a single steering vector $h(c_i^T) - h(c_i^F)$ is used as a steering vector.

This explains why the choice of mean of the activation differences across counterfactual pairs is a reasonable choice of steering vector. This is precisely the technique used in ITI. While they also experiment with other steering vectors, they found that this works the best for their experiments.

Now, we will continue on our insights to analyze whether we can build better steering vectors $\eta$. We present two crucial insights based on our analysis so far.

1. Looking at our desired equations, any *weighted combination* of $\eta_i = h(c_i^T) - h(c_i^F)$ will satisfy $Ah(x) = b_0, A(h(x) + \eta) = b_1$ exactly.

2. We could potentially choose the steering vector $\eta$ to be a function of $x$ instead of being a constant vector, provided $\eta(x)$ is efficiently computable during inference time.

Exploiting our first insight, we conclude that choosing any weighted combination of the $\eta_i$ should be a reasonable choice of steering vector provided we can control its effects on the spaces orthogonal to $A$. That is, we can choose

$$\eta = \sum_i w_i \eta_i = \sum_i w_i(h(c_i^T) - h(c_i^F))$$

as our steering vector. This gives us the extra freedom to tune the weights $w_1, w_2, \ldots$ based on other heuristics. Note that this also captures the choice of the top principal component of the steering vector as experimented in [118].

Our second observation suggests that even the steering vector $\eta$ could be a function of $x$, namely $\eta(x)$, provided it's efficiently computable during inference. Therefore, this suggests the usage of

$$\eta(x) = \sum_i w_i(x)(h(c_i^T) - h(c_i^F))$$

as our steering vector where the weights $w_i(x)$ depend on $x$.

Based on these two observations, we propose our ITI modification. We choose the steering vector to be dependent on the context $x$, with weights chosen to be $w_i = \langle \lambda(x), \lambda(c_i^F) \rangle$ for a sentence embedding $\lambda$ (such as Sentence-BERT [97]). That is,

$$\eta(x) = \sum_i \langle \lambda(x), \lambda(c_i^F) \rangle (h(c_i^T) - h(c_i^F))$$

Indeed, this is reasonable as if a context $x$ is close to $c_i^F$ for a specific training sample $i$ in terms of their sentence embeddings $\lambda(x)$ and $\lambda(c_i^F)$, then this particular sample's steering vector should be upsampled. In other words, we can think of the training sample contexts as voting on their respective counterfactual steering vector, with weights determined by the similarity between the representation

of the test context and the representation of the sample context. A justification would be that $B(x)$ (the relevant concepts for a datapoint) depend smoothly on $x$ (proximity is measured by similarity of embeddings) so it makes sense to upweight close points to enforce that $x$ preserves similar concepts.

Finally, we need to argue that we can compute this efficiently during inference. For this, we exploit the structure of our steering vector representation as follows.

$$\eta(x) = \sum_i \langle \lambda(x), \lambda(c_i^F) \rangle (h(c_i^T) - h(c_i^F))$$

$$= \left( \sum_i (h(c_i^T) - h(c_i^F)) \lambda(c_i^F)' \right) h(x)$$

$$= M h(x)$$

for the matrix $M = \sum_i (h(c_i^T) - h(c_i^F)) \lambda(c_i^F)'$, where $v'$ denotes the tranposed vector. We remark that the weights $w_i(x)$ as used could potentially be negative but this is not an issue since the projection of the corresponding counterfactual vector in the direction of $B$ is still random and we finally normalize $\eta(x)$, so the magnitude doesn't matter.

Therefore, this steering can be done efficiently by precomputing the *steering matrix $M$* and then during inference, we simply compute the steering vector $\eta(x)$ as $\eta(x) = M h(x)$.

In Table 6, we show the results of our experiments with steering matrices. We use the open-source large language model LLaMA [119] with 7 billion parameters (open sourced version from Hugging Face) and the sentence transformer SBERT [97] for the sentence embedding. We report the accuracy of the multiple-choice track of TruthfulQA [66] over 3 random seeds and also the Cross-Entropy Loss and

Table 6: Comparison of steering vectors for LLM alignment

| Technique | $\alpha$ | Acc. | CE loss | KL div. |
|---|---|---|---|---|
| Baseline | - | 0.257 ±0.00005 | 2.16±0.02 | 0.0±0.00 |
| Random direction | 20 | 0.258±0.002 | 2.19 ±0.02 | 0.02 ±0.002 |
| CCS direction | 5 | 0.262 | 2.21 | 0.06 |
| ITI: Probe weight dir. | 15 | 0.270 ±0.004 | 2.21 ±0.02 | 0.06 ±0.005 |
| ITI: Mass mean shift | 20 | 0.288 ±0.004 | 2.41 ±0.08 | 0.27±0.007 |
| Steering matrices (ours) | 15 | 0.295 ±0.02 | 2.61 ±0.07 | 0.41 ±0.04 |

KL divergence of the model pre- and post-intervention. All hyperparameters are tuned as per [66] and the experiments are performed on eight A6000 GPUs. Higher accuracy is better and lower CE loss, and KL divergence indicate that the original model has not been significantly modified. Here, the baselines are the unmodified model, random direction intervention, Contrast-Consistent Search (CCS) direction [17] and two different direction choices using vanilla ITI; and 2-fold cross validation is used.

We see that the multiple-choice accuracy improved, showcasing the potential of our steering matrices technique which is novel in the field of LLM alignment to the best of our knowledge. This is meant to be a proof of concept and not meant to be a comprehensive study of this specific technique. For exploratory purposes, we outline potential modifications to our technique below, which could potentially improve the performance, both in terms of accuracy as well as in terms of invasiveness. These form an exciting direction for a more comprehensive study of our proposed ideas, which we leave for future work.

**Implementation considerations**  We briefly note down some design choices we made in our implementation of the above method.

1. Since $\eta(x)$ is a function of $x$, the standard deviation of the activation projection on this direction, i.e., $\sigma_l^h(x)$ cannot be precomputed (as Li et al. [66] do), therefore we compute them dynamically during inference, which takes little overhead with fast tensorization operations (in particular, this is not the slow step).

2. We opted to go with evaluating the model only on the multiple-choice questions. This is partly because to evaluate the generated text, the recommended method is to use fine-tuned GPT-3-13B models but OpenAI have retired many of their older models as of this year, and therefore, the entire batch of experiments would have to be rerun with their newer models which could potentially change the baselines, and also because this work is a proof-of-concept rather than a comprehensive evaluation.

3. For computing the sentence embeddings, we only use the question prompts, as they contain all relevant contexts. And we normalize $\eta(x)$ during inference time.

**Additional ideas for improvement**    We re-iterate that our experimental exploration is not exhaustive and the preliminary experiments are merely meant to be a proof-of-concept. In this section, building on our insights, we outline some further ideas to improve the performance of ITI. We leave to future work to comprehensively explore these techniques in order to extract better performance towards LLM alignment.

1. Note that we opted to go with the weights $\langle \lambda(x), \lambda(c_i^F) \rangle$ where $\lambda$ was chosen to be a sentence transformer embedding [97]. While this is a reasonable choice, similarity metrics could be measured in other ways, e.g., with other sentence embedding models.

2. Going further, the weights do not have to be similarity scores and could be chosen via other heuristics. For instance, they could be chosen to be constants but potentially be optimized using a hold-out test set.

3. As Li et al. [66] noted, the ITI technique could be applied on top of fine-tuned models in order to further improve their performance. Therefore, our proposed modification could also potentially be applied on top of fine-tuned models.

# F    Contrastive algorithm for end-to-end concept learning

In this section, we present an end-to-end framework based on contrastive learning to learn the nonlinearity as well as concepts from data. This is inspired by the methods of the CRL work [14]. The model architecture is designed based on our concept conditional distribution parametrization. The core idea is as follows. For each concept conditional distribution $X^e$, we train a neural network to distinguish concept samples $x \sim X^e$ from base samples $x \sim X^0$. In Lemma 3, we derive the log-odds for this problem. Then, to learn the $n$ atomic concepts up to linearity, we build a neural architecture for this classification problem with the final layer mimicking the log-odds expression above, which can then be trained end-to-end. Because of the careful parametrization of the last layer, this will encourage the model to learn the representations as guaranteed by our results.

First, we will derive the computation of the true log-odds.

**Lemma 3.** *For any concept index $e$, there exist some constants $c_e$ such that*

$$\ln(p^e(Z)) - \ln(p(Z)) = \sum_{i=1}^{n} \left( -\frac{1}{2} M_{ei} \langle a_i, Z^e \rangle^2 + B_{ei} \langle a_i, Z^e \rangle \right) + c_e \tag{86}$$

*where $M, B$ are the environment-concept matrix and the environment-valuation matrix defined in* (7) *and* (8).

*Proof.* This follows from Eq. (15) in the proof of Theorem 2.    □

From our main identifiability results, we can assume without loss of generality that the concept vectors we learn are coordinate vectors. In other words, we consider a neural network $h^\theta$ with parameters $\theta$ with output neurons $h_1^\theta, \ldots, h_n^\theta$ such that the $n$ atomic concepts will now correspond to the concept vectors $e_1, \ldots, e_n$ (which is reasonable as they are only identifiable up to linear transformations). Therefore, for each environment $e$, we can train classifiers of the form

$$g_e(X, \alpha^e, \beta_k^e, \gamma_k^e, \theta) = \alpha^e - \sum_{k=1}^{n} \beta_e^k (h_k^\theta(X))^2 + \sum_{k=1}^{n} \gamma_e^k h_k^\theta(X) \tag{87}$$

equipped with standard cross-entropy loss, for hyperparameters $\alpha^e, \beta_k^e, \gamma_k^e, \theta$. Indeed, this is reasonable since if the training reaches the global optima in the ideal case, then the loss function will correspond to the Bayes optimal classifier and therefore, $g_e(X, \alpha^e, \beta_k^e, \gamma_k^e, \theta) = \ln(p^e(Z)) - \ln(p(Z))$, which along with Lemma 3 will suggest that the learnt network $h$ is linearly related to the function $A^e f^{-1}$, as desired. Lastly, we choose the loss function to be the aggregated CE loss and an extra

regularization term. That is,

$$\mathcal{L} = \sum_e \underbrace{-\mathbb{E}_{j\sim\text{Unif}(\{0,e\})}\mathbb{E}_{X\sim X^e}\left(\ln\frac{e^{\mathbf{1}_{j=e}g_e(X)}}{1+e^{g_e(X)}}\right)}_{\text{CE loss for environment } e} \quad + \quad \eta\|\beta\|_1 \tag{88}$$

for a regularization hyperparameter $\eta$.

**Sampling from concept conditional distributions**  A common task in controllable generative modeling is being able to generate data from a known concept. Note that this is not straightforward in our setting because the normalization term in Eq. (2) is not efficiently computable. To do this efficiently, we also outline a simple algorithm (Algorithm 1 in Appendix H) to sample from the concept conditional distribution for a known concept. Our proposed algorithm is based on rejection sampling and the algorithm as well as the complexity analysis is deferred to Appendix H.

## G  Additional details about the synthetic setup

In this section, we detail the synthetic setup in Section 6. The base distribution is sampled from a Gaussian mixture model with 3 components whose parameters are chosen randomly. The weights are randomly chosen from $\text{Unif}(0.3, 1)$ (and then normalized), the entries of the means are chosen from $\text{Unif}(-1, 1)$ and the covariance is chosen to be a diagonal matrix with entries in $\text{Unif}(0.01, 0.015)$ (note that the diagonal nature doesn't really matter since a map $f$ will be applied to this distribution). The mixing function $f$ is chosen to be either (i) linear or (ii) nonlinear with a 1-layer MLP containing 16 hidden neurons and $\text{LeakyReLU}(0.2)$ activations.

The number of concepts $n$ is intentionally chosen to be less than the ground truth dimension $d_z$ and the number of concepts is $m = n + 1$ as per our theory. The concepts are taken to be atomic, with the concept vectors and valuations chosen randomly, where each entry of the concept vector is chosen i.i.d from $\text{Unif}(-0.3, 0.3)$, and the resampling distribution is chosen to be a Gaussian with variance 0.005. Finally, we choose 5000 samples per environment, sampled via the rejection sampling Algorithm 1. For the contrastive algorithm, we choose the architecture to either be linear or nonlinear with a 2-layer MLP with 32 hidden neurons in each layer, with the final parametric layer chosen based on the known concept, to have the form described above. We train for 100 epochs, on a single A6000 GPU, with $\eta = 0.0001$ and use Adam optimizer with learning rates 0.5 for the parametric layer and 0.005 for the non-parametric layer, with a Cosine Annealing schedule [72].

Additional results for higher dimensional settings can be found in Table 7.

Table 7: Linear identifiability of synthetic settings, averaged over 5 seeds. Same setup as in Table 1 with larger values of $d_x$ and $d_z$.

| Mixing ($f$) | $(n, d_z, d_x)$ | $R^2\uparrow$ | MCC$\uparrow$ |
|---|---|---|---|
| Linear | (4, 15, 18) | $0.98 \pm 0.01$ | $0.98 \pm 0.03$ |
| Nonlinear | (4, 15, 18) | $0.89 \pm 0.04$ | $0.84 \pm 0.08$ |
| Linear | (4, 20, 22) | $0.94 \pm 0.04$ | $0.80 \pm 0.06$ |
| Nonlinear | (4, 20, 22) | $0.93 \pm 0.03$ | $0.84 \pm 0.08$ |
| Linear | (4, 25, 28) | $0.85 \pm 0.03$ | $0.79 \pm 0.07$ |
| Nonlinear | (4, 25, 28) | $0.76 \pm 0.11$ | $0.72 \pm 0.15$ |

## H  Controllable generative modeling via rejection sampling

In this section, we will describe how to sample from a concept conditional distribution with a known concept. Once the concepts are learned in our framework, we can use this technique to generate new data satisfying various desired concepts, which will aid in controllable generative modeling.

Consider the base distribution on $Z \in \mathbb{R}^{d_z}$ with density $p(Z)$. Suppose we wish to sample from a concept $C$ given by $AZ = b$ and resampling distribution $q$. We additionally assume that $q$ is efficiently computable and an upper bound $L$ is known for its density, i.e., $L \geq \max(q)$.

Recall that the desired density is defined as

$$p_C(Z) \propto p(Z) \prod_{i \leq dim(C)} q((AZ - b)_i)$$

Note that it's infeasible to compute the normalization constant for such complex distributions. However, we bypass this by using rejection sampling. We describe the procedure in Algorithm 1.

---

**Algorithm 1:** Rejection sampling for controllable generative modeling

---

**Input:**

- Base distribution $p$
- Resampling distribution $q$ with upper bound $L \geq \max(q)$
- Concept $C$ with transformation $A$ and valuation $C$

**Output:** Returns a single sample from $p_C(Z)$

1  $M = L^{dim(C)}$
   // Repeat trials until condition is met
2  **while** *True* **do**
3  $\quad$ Z = yield(p)
4  $\quad$ U = yield(Unif(0, 1))
5  $\quad$ $R = \frac{1}{M} \prod_{i \leq dim(C)} q((AZ - b)_i)$
6  $\quad$ **if** $R \geq U$ **then**
7  $\quad\quad$ **return** $Z$

---

Informally, we first sample $Z \sim p$ (we overload notation for both density and the distribution) and an independent variable $U \sim Unif(0, 1)$, the uniform distribution on $(0, 1)$. We accept the variable $Z$ if

$$\frac{1}{M} \prod_{i \leq dim(C)} q((AZ - b)_i) \geq U$$

for a predetermined upper bound $M$ on the quantity $\prod_{i \leq dim(C)} q((AZ - b)_i)$. If the inequality is false, we simply reject the sample and repeat.

Now we will argue why this algorithm is correct, which is accomplished in Theorem 3. Let

$$N_C = \int_Z p(Z) \prod_{i \leq dim(C)} q((AZ - b)_i)$$

be the normalization constant in the definition of $p_C(Z)$. Therefore

$$p_C(Z) = \frac{1}{N_C} p(Z) \prod_{i \leq dim(C)} q((AZ - b)_i)$$

**Lemma 4.** *Let $M \geq \max(q)^{dim(C)}$ The acceptance probability of each iteration of the while loop in Algorithm 1 is $Pr[Z \text{ accepted}] = \frac{N_C}{M}$*

*Proof.* We have

$$Pr[Z \text{ accepted}] = Pr_{U,Z} \left[ U \leq \frac{1}{M} \prod_{i \leq dim(C)} q((AZ - b)_i) \right]$$

$$= Pr_{U,Z} \left[ U \leq \prod_{i \leq dim(C)} \frac{q((AZ - b)_i)}{\max(q)} \right] \qquad \text{since } M \geq max(q)^{dim(C)}$$

$$= \int_Z Pr_U \left[ U \leq \prod_{i \leq dim(C)} \frac{q((AZ - b)_i)}{\max(q)} \right] p(Z) \, dZ \quad \text{as } U, Z \text{ are independent}$$

$$= \int_Z \left[ \prod_{i \leq dim(C)} \frac{q((AZ - b)_i)}{\max(q)} \right] p(Z) \, dZ \qquad \text{since } \frac{q((AZ - b)_i)}{\max(q)} \leq 1 \text{ always}$$

$$= \int_Z \frac{N_C p_C(Z)}{M} \, dZ$$

$$= \frac{N_C}{M}$$

$\square$

Before we prove correctness, we will remark on the expected number of trials needed for accepting each sample.

**Corollary 1.** *The expected number of trials needed to generate a single sample is* $\frac{M}{N_C}$

*Proof.* Note that each iteration of the while loop is independent, therefore the number of trials until acceptance is distributed as a geometric random variable whose expectation is the inverse of the parameter. $\square$

This suggests that for our algorithm to be efficient in practice, $M$ should be chosen as small as possible, i.e., estimates of $\max(q)$ should be as tight as possible.

**Theorem 3.** *Algorithm 1 yields samples from the concept conditional distribution* $p_C$.

*Proof.* The proof is at heart the proof of correctness of rejection sampling. For arbitrary parameters $t_1, \ldots, t_{d_z} \in \mathbb{R}$, let's compute the cumulative density of the samples output by Algorithm 1 and show that it matches the cumulative distribution function of $p_C(Z)$ evaluated at $t_1, \ldots, t_{d_z}$, which will complete the proof. That is, we wish to calculate

$$Pr[Z_1 \leq t_1, \ldots, Z_{d_z} \leq t_{d_z} | Z \text{ accepted}] = \frac{Pr[Z_1 \leq t_1, \ldots, Z_{d_z} \leq t_{d_z}, Z \text{ accepted}]}{Pr[Z \text{ accepted}]}$$

We already computed the denominator in Lemma 4. Therefore,

$$Pr[Z_1 \leq t_1, \ldots, Z_{d_z} \leq t_{d_z} | Z \text{ accepted}]$$

$$= \frac{M}{N_C} Pr[Z_1 \leq t_1, \ldots, Z_{d_z} \leq t_{d_z}, Z \text{ accepted}]$$

$$= \frac{M}{N_C} \mathbb{E}_Z \left[ \mathbb{1}_{Z_1 \leq t_1} \ldots \mathbb{1}_{Z_{d_z} \leq t_{d_z}} \cdot \mathbb{E}_U [\mathbb{1}_{Z \text{ accepted}}] \right]$$

$$= \frac{M}{N_C} \mathbb{E}_Z \left[ \mathbb{1}_{Z_1 \leq t_1} \ldots \mathbb{1}_{Z_{d_z} \leq t_{d_z}} \cdot \frac{1}{M} \prod_{i \leq dim(C)} q((AZ - b)_i) \right] \qquad \text{from the proof of Lemma 4}$$

$$= \int_Z \mathbb{1}_{Z_1 \leq t_1} \ldots \mathbb{1}_{Z_{d_z} \leq t_{d_z}} \cdot \frac{1}{N_C} \prod_{i \leq dim(C)} q((AZ - b)_i) p(Z) \, dZ$$

$$= \int_Z \mathbb{1}_{Z_1 \leq t_1} \ldots \mathbb{1}_{Z_{d_z} \leq t_{d_z}} \cdot p_C(Z) \, dZ$$

which is precisely the cumulative distribution function of $p_C(Z)$ evaluated at $t_1, \ldots, t_{d_z}$. $\square$

