# OpenReview forum: "From Causal to Concept-Based Representation Learning"
_NeurIPS.cc/2024/Conference — NeurIPS 2024 poster_

### Official Review · Reviewer_kVe2 · 2024-07-08

**Soundness:** 3
**Presentation:** 2
**Contribution:** 3
**Rating:** 6
**Confidence:** 2

**Summary:**

The paper focus on recover human-interpretable concepts from observation. It proposes a concept based representation learning method, which relax causal notions with a geometric notion of concepts. Experiments on synthetic data, multimodal CLIP models and  large language models supplement their results and show the utility of their approach.

**Strengths:**

1.The authors hope to find a middle ground where they can simultaneously identify a smaller set of interpretable latent representations, which is an interesting idea.
2.This work can be interpreted as a new direction for identifiable representation learning in order to study when interpretable concepts
can be recovered from data.
3.Experiments on synthetic data, multimodal CLIP models and large language models supplement the results and show the utility of the approach.

**Weaknesses:**

1.This approach sacrifices causal semantics. This can be particularly problematic in situations where a deep understanding of causality is crucial, such as in root cause analysis, where the goal is to identify the fundamental reasons behind a problem or an event. Without causal semantics, one might only address the symptoms rather than the core issues, leading to temporary or ineffective solutions.
2.The author has made numerous assumptions within the article, which could potentially impact the universal applicability of the theory presented.

**Questions:**

Please see the weaknesses.

**Limitations:**

The authors adequately addressed the limitations.

---

> ### Author Rebuttal · Authors · 2024-08-07
>
> We thank the reviewer for their review. We address their comments below.
>
> > This approach sacrifices causal semantics. This can be particularly problematic in situations where a deep understanding of causality is crucial
>
> In this work, we focus on the many applications where a relaxation of causal semantics is allowed or necessary, such as when interventional data is out of reach (L46-50, L118-124).
> While a deep understanding of causality would be ideal, concept learning is itself a mature field independent of causality and we present a middle ground that is useful in contemporary applications.
>
> > The author has made numerous assumptions within the article, which could potentially impact the universal applicability of the theory presented.
>
> We would like to highlight that without any assumptions, we cannot have identifiability (L29-31),
> therefore we make natural assumptions to prove our results.
> However, we tried to keep the assumptions as general as possible, e.g. we allow non-linear mixing (Assumption 1) and use Gaussian noise (Assumption 3).
>
> We welcome additional suggestions to improve the work.

---

### Official Review · Reviewer_ohha · 2024-07-13

**Soundness:** 4
**Presentation:** 2
**Contribution:** 3
**Rating:** 6
**Confidence:** 4

**Summary:**

This paper proposes a theory to identify latent concepts from model representations. In contrast to previous work in the concept-based models' field, concepts are expected to be linearly represented in the model representations, and a linear transformation A is associated with such concepts. The paper is theory-driven and presents rigorous results for the identifiability of such concepts. The authors also present some experiments related to the theory, one of which verifies in a synthetic setup the validity of the proposed theory, one on CLIP and one on LLMs.

**Strengths:**

### **Originality**

The paper proposes a new perspective on concept-based learning by providing also identifiability guarantees. The notion of concepts as linear subspaces in the representation space is interesting although not entirely new, see [1]. The results on identifiability are new and potentially useful for follow-up works: it is a valid idea to learn such concepts from conditional distributions.

### **Quality and Clarity**

The paper is of high quality, providing new important results for identifiability of latent concepts, and addresses the important problem of learning high-level concepts from data. The presentation is somehow clear, although it can be improved.

### **Significance**

Bridging identifiability in (causal) representation learning and concept-based learning is an open problem. The main contribution by the authors is indicating a viable route to achieve it by relating with theory. It also can be related to visual-language models, like CLIP, that are typically learned without any concept supervision but, due to interpretability concerns, it is often of interest to know whether concepts are linearly represented and used by the model. The theory seems solid and is of interest for advancing research in concept-based models.


[1] Concept Whitening for Interpretable Image Recognition, Chen et al. (2020)

**Weaknesses:**

## Major

### **Experiments**

One thing that is particularly dangling in the paper is the experimental section and the supposed evidence in support of the theory:

1) The experiments on CLIP and LLMs seem to me unrelated to the theory devised by the authors and rather support on one hand that concepts can be found in the representations in CLIP to some extent (which was previously observed, e.g. [2,3]), on the other hand, that it is possible to improve the steering of LLMs predictions with some matrix operation rather than vector addition. How is this related to the theory the authors propose? How is it that CLIP training aligns with the assumptions that would lead to the identifiability of latent concepts? LLMs like LLaMA or GPT are next-token predictors learning with a different objective, see [4], how are even these models related to the theory proposed?

2) It would have been more useful to provide experimental evidence of the proposed contrastive learning method on semi-synthetic datasets like MPI3D or Shapes3D.
 Some real-world datasets are of particular interest to the community in concept-based models, being more tackling, like CUB200 and OAI [5], CelebA [6], and many more [7,8].

3) The synthetic experiment is impossible to understand from the main text and many details concerning the data, the learning procedure proposed (which is also very detailed), and the metrics are confined to appendices rendering it necessary to consult them in length. Nonetheless, it seems that under the working assumptions the model trained in a contrastive manner on the synthetic data and environments captures the right latent concepts. How do you evaluate $A^e$?

### **Assumptions**

It would be beneficial to present the assumptions and discuss the intuition behind them. Assumption 1 is fine and is common in studying identifiability, but if the aim is to identify concepts, it seems unnaturally restricting to consider only invertible functions $f$. One could hope to extend the results also to non-invertible functions. However, this is not a serious limitation given the novelty of the results.  \
Assumption 2 takes the concepts to be linearly independent, what happens if for some of them is not the case? \
Assumption 3 requires a Gaussian distribution for the noise, it is remarked that other distributions work as well (which distributions?) but there is no citation. \
How should Assumptions 4 and 5 be understood, what data are expected to be collected? Are these expensive to obtain in practice? They seem to presuppose a lot of knowledge on what latent concepts should be identified, which may not be the case when concepts are not known apriori.


## Minor

### **Conditional distributions in practice**

It is a bit puzzling how data should be gathered for the theory to work, and thus the scaling of the proposed method. The synthetic experiment offers a proof of principle but it does not show how the model behaves in settings where more concepts (and structured ones, like the color of an object) and conditional distribution are to be considered.


### **Related work**

Causal Representation Learning seems more of an inspiration to the paper rather than having a tight connection to the theory. The authors consider conditional distributions for latent concepts and the proof techniques are inspired by the iVAE work (2019), so there is no clear link to what should be the causal aspects that should be taken into consideration.

It seems more that works about identifiability in Causal Representation Learning become relevant if one wants to extend authors' theory to causal variables, not being essential to support their claims and the connection.

I was also expecting to see a comparison to Concept-based models' current practices, which require dense supervision of the concepts in cases [1, 7, 8],  partial [6], or language-guided [2,3]. Some approaches aim to learn concepts only by leveraging supervision on the classification task, see [9], and seemingly related work on continuous latent variable identification [10].

## Summary

The experiments do not complement the theory and dilute the message by showing two post-hoc analyses on CLIP and LLMs. I struggle to see how both constitute valid evidence for the theory proposed. On the other hand, additional experiments on known semi-synthetic datasets or real ones would highlight the extent of the theory to the community in concept-based interpretability.

The presentation of the material also requires some clarification around the assumptions the authors make and their validity in practice.

[2] Label-free Concept Bottleneck Models, Oikarinen et al. (2023) \
[3] Language in a Bottle: Language Model Guided Concept Bottlenecks for Interpretable Image Classification, Yang et al. (2023) \
[4] On Linear Identifiability of Learned Representations, Roeder et al. (2021) \
[5] Concept Bottleneck Models, Koh et al. (2020) \
[6] GlanceNets: Interpretable, Leak-proof Concept-based Models, Marconato et al. (2022)  \
[7] Concept embedding analysis: A review, Schwalbe (2022) \
[8] Concept-based explainable artificial intelligence: A survey, Poeta et al. (2023) \
[9] Provable concept learning for interpretable predictions using variational autoencoders, Taeb et al., (2022) \
[10] Synergies between Disentanglement and Sparsity: Generalization and Identifiability in Multi-Task Learning, Lachapelle (2023)

**Questions:**

**Recovering the concepts without post-hoc analysis?**
As I understood correctly (correct if I am wrong), the identifiability class studied in the paper presupposes recovering the latent concepts (the matrix A) up to an invertible transformation $T$. Thus, the theory offers only guarantees to which a linear probe on the latent representations would recover the latent concepts, is it correct? One in practice has still to find the matrix $A$ related to the concept, and that cannot be done without concept annotation. Is that the case?

**Typo?** In Def  It seems that $d_Z$ and $\tilde{d_Z}$ have to be the same to guarantee the inverse exists (and how it is used in the proofs). Is that the case? Would the theory hold also for models with different latent dimensions $d_Z \neq \tilde{d_Z}$?

I asked other questions in the weakness part.

**Limitations:**

All assumptions are natural limitations of the proposed theoretical results. It is not clear whether foundation models are trained under the conditions that the author found for assessing identifiability of the concepts
and the connection to LLMs seems a bit weak.

---

> ### Author Rebuttal · Authors · 2024-08-07
>
> We thank the reviewer for their careful review and are glad that they appreciate our contributions towards the important open problem of bridging causal representation learning and concept based representation learning.
>
> > The experiments on CLIP and LLMs seem to me unrelated to the theory...How is this related to the theory the authors propose?
>
> The CLIP experiments showcase
> that the learned representations
> of concepts indeed lie approximately within an affine hyperplane which is an illustration of our key hypothesis.
> Moreover, the training data approximately satisfies our assumptions if we view the caption as the environment, i.e., images with similar/equal captions form a concept conditional distribution and the joint image distribution is the observational distribution.
> Our experiments also show that  the representations of different CLIP models agree up to linear transformation.
> As noted by the reviewer,
> a more rigorous connection would be to establish if CLIP models learn a representation such that concepts are represented by affine hyperplanes and whether concept conditional distributions are approximately given by eq. (2).
> In such a case, we can apply our identifiability theory
> to conclude that different CLIP models necessarily learn approximately the same representation
> up to linear transformation.
> Proving these facts rigorously for the representation based on the CLIP-loss is a challenging problem we leave for future work.
>
> The LLM experiment is a bit more exploratory and the connection to our theory is via our conceptualisation of concepts. Indeed, this helps us to obtain intuition what the right steering vector should look like and guided our construction.
>
> > It would have been more useful to provide experimental evidence ... on semi-synthetic datasets like MPI3D or Shapes3D. Some real-world datasets ... like CUB200 and OAI [5], CelebA [6], and many more [7,8].
>
> We appreciate the nice suggestions. Since the primary thrust of the work is theoretical and it is currently quite challenging to scale these type of methods to large datasets, we leave this interesting direction for future work.
>
> > The synthetic experiment is impossible to understand from the main text and many details ... are confined to appendices...How do you evaluate $A^e$?
>
> We apologize that this part is difficult to understand as details were deferred due to lack of space.
> In order to evaluate $A^e$ in our synthetic experiments, we compute the linear correlation metrics $R^2$ and MCC between the ground truth $Z$ (restricted to the projection space) and the predicted $Z$ (from our model), which we report in our tables.
> As outlined to other reviewers, we will use additional space to add experimental details to the main text.
>
> About assumptions in general, we believe our assumptions could likely be further relaxed at the expense of more technical work, which we leave for future work.
>
> > Assumption 1 is fine and is common in studying identifiability. One could hope to extend the results also to non-invertible functions. However, this is not a serious limitation given the novelty of the results.
>
> We agree.
>
> > Assumption 2 takes the concepts to be linearly independent, what happens if for some of them is not the case?
>
> Then we can in general not identify
> the concept matrix, think, e.g., of the case where two concepts are collinear.
>
> > Assumption 3 requires a Gaussian distribution for the noise, it is remarked that other distributions work as well (which distributions?) but there is no citation.
>
> The result could be extended to exponential families, we will clarify this and add a reference (see e.g. Khemakhem et al. [50]).
>
> > How should Assumptions 4 and 5 be understood, what data are expected to be collected?...how data should be gathered for the theory to work, and thus the scaling of the proposed method.
>
> Assumptions 4 and 5 state that the environments need to be sufficiently diverse. We do not think that the approach should be used to collect data in practice because this would indeed require a lot of prior knowledge which could probably be directly used to learn the concepts. Instead we think that often real-world data comes with subtle heterogeneity that allows us to identify  concepts.
>
> > Causal Representation Learning seems more of an inspiration to the paper... not being essential to support their claims and the connection.
>
> We generally agree with the reviewer
> and refer, e.g., to the discussions
> in L43-50, 60-63, for this remark. We're happy to add more details to clarify this.
>
> > I was also expecting to see a comparison to Concept-based models' current practices
>
> We appreciate the relevant references and are happy to include them in the paper. In short, some of these works study very useful empirical methods whereas we focus on rigorous theoretical contributions, whereas the others differ in the kind of assumptions and settings studied, i.e. they're related but not directly comparable at a technical level.
>
> > Recovering the concepts without post-hoc analysis?...One in practice has still to find the matrix $A$ related to the concept, and that cannot be done without concept annotation. Is that the case?
>
> Yes, our identifiability is only upto linear transformations. After having learned the nonlinearity, in order to recover the transformation itself, additional information such as concept annotation may be utilized in practice.
>
> > It seems that $d_Z$ and $\tilde{d_Z}$ have to be the same to guarantee the inverse exists (and how it is used in the proofs). Is that the case? Would the theory hold also for models with different latent dimensions $d_Z \neq \tilde{d_Z}$?
>
> Yes, for technical reasons this is the case.  To consider $d_Z\neq \tilde{d_Z}$ we need to relax the injectivity assumptions otherwise $d_Z$ has to correspond to the dimension of the data manifold.
>
> We hope the rebuttal answered the questions raised, and are also happy to take additional feedback to improve the writing.

---

> ### Comment · Reviewer_ohha · 2024-08-08
> **Reply to authors**
>
> Thank you for your clarifications. After reading the reply and other reviews I feel some parts of the paper still present the limitations we pointed out (experimental section).  I have some comments for your rebuttal:
>
> > We appreciate the nice suggestions. Since the primary thrust of the work is theoretical and it is currently quite challenging to scale these type of methods to large datasets, we leave this interesting direction for future work.
>
> I understand that scaling to real-world datasets is challenging and a bit out of scope. However, it should be shown that the method can applied to semisynthetic datasets (even MNIST, which is 28x28 dimensional), otherwise leaving me puzzled about the practical utility of the theory and method proposed. Given the theoretical focus of the authors, I accept their decision to do that in future work.
>
> > The CLIP experiments showcase that the learned representations of concepts indeed lie approximately within an affine hyperplane which is an illustration of our key hypothesis.
>
> This is ok. It seems to me more of a verification of an interesting empirical fact rather than a consequence or an explanation that is already captured by the theory.
>
> > In such a case, we can apply our identifiability theory to conclude that different CLIP models necessarily learn approximately the same representation up to linear transformation.  Proving these facts rigorously for the representation based on the CLIP-loss is a challenging problem we leave for future work.
>
> That would be interesting, but I would expect the theory would require several adjustments to consider the captions to train visual representations.
>
> > The LLM experiment is a bit more exploratory and the connection to our theory is via our conceptualisation of concepts. Indeed, this helps us to obtain intuition what the right steering vector should look like and guided our construction.
>
> Similarly, this is an interesting empirical fact, but not explained by the theory.
>
> > As outlined to other reviewers, we will use additional space to add experimental details to the main text.
>
> Thank you.
>
> > We do not think that the approach should be used to collect data in practice because this would indeed require a lot of prior knowledge which could probably be directly used to learn the concepts. Instead we think that often real-world data comes with subtle heterogeneity that allows us to identify concepts.
>
> What do you mean by real-world data having this heterogeneity, is it something that could be verified? It seems to me that you have first to assume what are your concepts of interest (being linearly related to generative variables) and then to have those environments. Both points seem challenging.
>
> > We generally agree with the reviewer and refer, e.g., to the discussions in L43-50, 60-63, for this remark. We're happy to add more details to clarify this.
>
> Thank you.
>
> > In short, some of these works study very useful empirical methods whereas we focus on rigorous theoretical contributions, whereas the others differ in the kind of assumptions and settings studied, i.e. they're related but not directly comparable at a technical level.
>
> I also agree with this. Common practice has been shifting towards using concept-supervised examples (which trivially address the identifiability of concepts) which is a limitation and your theory is very relevant in relaxing it.
>
> > Yes, our identifiability is only up to linear transformations. After having learned the nonlinearity, in order to recover the transformation itself, additional information such as concept annotation may be utilized in practice.
>
> Thank you for the clarification. It would be interesting to connect to [1] to see if the same can be done without concept supervision.
>
> [1] When are Post-hoc Conceptual Explanations Identifiable? Leeman, UAI 2023.

---

> > ### Author Response · Authors · 2024-08-12
> >
> > Thanks for your comments concurring with the positioning of this paper in relation to the existing literature. We also appreciate your open-mindedness and willingness to allow some of the experimental concerns to be sorted out in more dedicated future work. And yes, the work [1] you cited is along those lines (we briefly highlight this work in L163-165).
> >
> > We again thank the reviewer for their great effort in reviewing the paper and are happy to take additional questions or suggestions.

---

### Official Review · Reviewer_jSX2 · 2024-07-19

**Soundness:** 3
**Presentation:** 3
**Contribution:** 4
**Rating:** 7
**Confidence:** 4

**Summary:**

This work takes a step toward learning human-interpretable concepts while relaxing the restrictions of (interventional) causal representation learning, and they do so inspired by the linear representation hypothesis.

The authors claim that learning the generative process and the "true" causal factors $f^{-1}, Z$ from observations $X$ using interventions has important caveats: 1) One needs many interventions ($\Omega(d_z)$) for identifiability of $f^{-1}, Z$ which might be too much of a requirement in many cases, 2) There's no reason that a priori such latent representations are interpretable, 3) Interventions in many examples might not be possible at all 4) One might not need the whole $f^{-1}, Z$, and there are cases where we can seek only a handful of interpretable concepts for an application without learning the full encoder and latent representation.

The authors then introduce the geometric notion of concepts as linear subspaces in the latent space of $Z$. This is inspired by the abundant evidence on the linear representation hypothesis. Based on this notion, they define the concept conditional distributions as a source of supervision for learning concepts which will replace interventional distributions as the source of supervision for learning causal representations $Z$. Concept conditional distributions are simply defined by filtering the dataset with samples that are $\textit{perceived}$ to satisfy a concept (see eq. 1)

The problem then becomes whether given an observational distribution $X^0$ and a set of concept-conditional distributions $X^1, \dots, X^m$ corresponding to $m$ concepts, one can identify the linear subspaces $A^mf^{-1}(x)$ corresponding to those concepts.

The main theorem then proves the identifiability (according to definition 4) of those concepts given linear independence of concepts, as well as some diversity constraints on the environments.

The authors then try to validate the claim using 3 experiments:
1) Synthetic experiments with various linear and non-linear mixing $f$, and different dimensions for $Z,X$.
2) Evaluating the linearity of the valuations of the concepts learned via multi-modal CLIP (inspired by the similarity of CLIP objective to their contrastive algorithm
3) Showing that concepts can be used to steer LLM outputs.

**Strengths:**

- I find this work original and novel. There have been attempts at learning concepts recently, but from my understanding (as well as the author's mention of the related work) such attempts have been limited to specific domains, while the work at hand seems to be addressing that challenge in a broad way.
- Moreover, I believe they have nicely translated the linear representation hypothesis to concept learning, and relaxing the restrictions of interventional causal representation learning is an important endeavor (see questions though) as is also nicely motivated in multiple places in the paper.
- I find the theoretical result insightful and important; not only does this work move away from interventions as a not-so-ideal tool, but also clearly demonstrates the theoretical advantage of concepts vs. interventions (only theoretically though).
- The experiments touch upon different modalities showing the versatility of the claims.
- The presentation and arguments are generally well-constructed (up until page 8)

**Weaknesses:**

- In the synthetic experiments, I was expecting to see large $d_z$ and small $n$ to match the claims that had been made earlier as to the advantage of concept conditional distributions, but the dimensions are quite small. I can see that they show a proof of concept, but still, it would have been nice to be consistent with the claims made earlier (unless there's a reason why the authors didn't do so)

- Could the authors think of any experiment to contrast concept learning to CRL? Maybe with simple datasets like CLEVR or 3d-shapes? Isn't there a way to try to learn causal representations and concepts, and show empirically that one is easier to be achieved? I understand that the premise was that $Z$ is not always interpretable in the first place, but I think there would exist situations where it will be. If I understand correctly, do the authors think such an experiment would add to the empirical evidence for their method?

- I'm also a bit unclear about where this work is taking us, and would have liked it if it was explained better. In particular, are we hoping to change our representation learning towards learning concepts? If so, are you proposing this for reasoning tasks? For alignment? Interpretability? Or what else? The reason I'm asking is that wouldn't we probably still need some causal representations in some reasoning tasks, say in vision? Basically a short discussion of when we would prefer concepts over causal representations (or else) would be helpful.

----
Writing and Clarity:

- The notion of environment in the context of concepts appeared out of the blue on page 7 (same with its notation that followed)
- Not a weakness of the method - but the learning method seems like an important component of the paper which is deferred altogether to the appendix, i.e., one would learn about the identifiability, but there is no mention of the actual method to learn such (identifiable) concepts which might hide the difficulties and challenges associated with it.



Please also see the questions.

**Questions:**

- Could you connect/contrast this work the recent advances in sparse autoencoders (SAEs)? While the problems are somewhat differently motivated, the resulting concepts bear resemblance to the learned sparse features from activations of transformers. Could the authors comment on this please?
- If we care about a few features, why not use CRL methods that guarantee weaker identification requiring much less interventions? (See question below).
- I generally agree with the motivation and direction of the paper, but I feel like the CRL community has been aware of these restrictions and recently took steps to address them; for instance, from what I understand the multi-node intervention line of work (cited by the authors) alleviates the challenge of perfect interventions and relaxes the requirement of learning all of $Z$, instead weaker and more general notions of identifiability have been introduced. A proper discussion on this would be helpful (more than what is in the appendix). Wouldn't it make sense to leverage such methods in tasks where we have a prior that there is some underlying causal representation?
- Does equation 2 come from the independence of concepts (asking because that is introduced later). Why is not $k=dim(C)$?
- Line 294 onwards, is $e$ properly defined before and used here?
- Line 304, is $S_n$ a typo? $n$ was used as superscript before that, here it's used as a subscript.
- Could assumption 4 be explained in words as well? Where does it come from? When/in what situations does it break?
- How should one interpret table 2 of the appendix? Is there a reason why it's not plotted, and presented as numbers? Shouldn't we expect a linear plot?
- Line 165, what do you mean by entangled concepts, is there an experiment to show that? Or do you mean the superposition of atomic concepts?
- (Not a question impacting the score) Related to independence of concepts: Can the theory say anything about hierarchical concepts similar to hierarchical representations?

Remark: I am willing to increase my score if some of the questions and weaknesses are discussed since I find the direction of this work quite interesting and important.

**Limitations:**

See questions.

---

> ### Author Rebuttal · Authors · 2024-08-07
>
> We thank the reviewer for their detailed review and nice summary of the paper. We also appreciate their insightful comments on the importance of this work.
>
> > In the synthetic experiments, I was expecting to see large $d_z$ and small $n$
>
> Thanks for the suggestion, we ran additional experiments and included the results in the global response. We kept $n$ as $4$ and increased $d_z$. While the metrics naturally degrade a bit, they're still comparable to other nonlinear CRL works.
>
> > Could the authors think of any experiment to contrast concept learning to CRL?
>
> This is an interesting suggestion and would potentially add to the empirical evidence. However, it is beyond the scope of our work to compare against CRL comprehensively, partly because we're studying settings where CRL is not directly applicable (e.g., due to lack of interventional data). Moreover, our work is primarily theoretical and we leave for future work to extensively compare concept learning methods to CRL techniques empirically.
>
> > I'm also a bit unclear about where this work is taking us, and would have liked it if it was explained better. In particular, are we hoping to change our representation learning towards learning concepts?
>
> Our hope with this work is to provide concept learning a rigorous footing via the theory of identifiability. We do not focus on a specific task and instead build a conceptual bridge between causal representations and concept learning.
> While causal representations may be ideal or even necessary for some tasks, our work presents a middle ground for many contemporary settings where this may not be possible (please see also L43-50).
> As reviewer `ohha` points out, "Bridging identifiability in (causal) representation learning and concept-based learning is an open problem" and this is one of the main motivations of our work.
> We will include this discussion in the paper.
>
> > The notion of environment in the context of concepts appeared out of the blue on page 7
>
> Thank you, we will make sure to introduce this terminology carefully.
>
> > Not a weakness of the method - but the learning method seems like an important component of the paper which is deferred altogether to the appendix
>
> We agree that the contrastive learning method is an important component of the work, however, we have regrettably deferred the details to Appendix F due to lack of space. With the additional page available in the final version, we are happy to include this in the main paper.
>
> We next address the questions in order.
>
> > Could you connect/contrast this work the recent advances in sparse autoencoders (SAEs)?
>
> We're briefly aware of works on SAEs that learn interpretable features in models. However, to the best of our knowledge, they do not provide theoretical identifiability guarantees for learning concepts, which our work endeavors.
>
> > If we care about a few features, why not use CRL methods that guarantee weaker identification requiring much less interventions?...the multi-node intervention line of work (cited by the authors) alleviates the challenge of perfect interventions and relaxes the requirement of learning all of $Z$
>
> Could the reviewer please clarify which specific work they mean? We're more than happy to comment further and have a proper discussion on this in the paper as well.
>
> > Does equation 2 come from the independence of concepts. Why is not $k = dim(C)$?
>
> Yes, equation 2 arises if the noise for the noisy estimates is independent for all atomic concepts.
> The product runs from 1 to $\mathrm{dim}(C)$ so our expression is the same as
> $\prod_{k=1}^{\mathrm{dim}(C)}$.
>
> > Line 294 onwards, is $e$ properly defined before and used here?
>
>  We will clarify that $e$ is just an environment label for a concept conditional distribution.
>
> > Line 304, is $S_n$ a typo?  $n$ was used as superscript before that, here it's used as a subscript.
>
> No, here $S_n$ corresponds to the permutation group on $n$ elements, we will use a different font to make the distinction clear.
>
> > Could assumption 4 be explained in words as well? Where does it come from? When/in what situations does it break?
>
> This condition is similar to other diversity conditions in identifiability theory. It ensures that there are sufficiently many, non-redundant datasets. It breaks, e.g.,
> if there are fewer datasets than atomic concepts of interest or when several of the concept conditional datasets disagree.
>
> > How should one interpret table 2 of the appendix? Is there a reason why it's not plotted, and presented as numbers? Shouldn't we expect a linear plot?
>
>  Note that for the hue variables the different numbers correspond to different colors but the numbers
>  do not correspond to meaningful concept valuations, but are just discrete labels for the different colors. Therefore, we do not expect
>  to recover a linear relation between the evaluated concept valuations and the label index.
>  This is different for attributes such as, e.g., size or orientation where the value should correspond approximately to the valuation (potentially up to a non-linear transformation).
>  Note that it is moreover not clear whether color is an atomic concept as we assume here (e.g., standard color representations are 2 dimensional).
> We nevertheless observe a high correlation coefficient between the representations learned by different models (see Table 5).
>
> > Line 165, what do you mean by entangled concepts...Related to independence of concepts: Can the theory say anything about hierarchical concepts similar to hierarchical representations?
>
> We would like to clarify that in assumption 2, we talk about linear independence (and not statistical independence) of atomic concepts. However, the concepts we actually allow can each consist of multiple atomic concepts and different non-atomic concepts can overlap (not superposition), which can be interpreted as hierarchical or entangled concepts as well.
>
> We thank the reviewer for their suggestions and welcome additional feedback to improve the text.

---

> > ### Comment · Reviewer_jSX2 · 2024-08-09
> >
> > Thank you for your efforts and for the clarifications.
> >
> > Also, thanks for carrying out experiments with larger $d_z$ and smaller $n$. Although I originally brought it up so there is an experiment aligning with the premise of the paper about very large $d_z$, I find this additional experiment fine as well.
> >
> > I appreciate the following comment
> > > This is an interesting suggestion and would potentially add to the empirical evidence. However, it is beyond the scope of our work to compare against CRL comprehensively, partly because we're studying settings where CRL is not directly applicable (e.g., due to lack of interventional data). Moreover, our work is primarily theoretical and we leave for future work to extensively compare concept learning methods to CRL techniques empirically.
> >
> > However, since the paper contrasts itself a number of times to CRL requiring many interventions, I don't see it totally beyond the scope of this work to make a minimal effort at contrasting to basic CRL methods. I agree with the authors that the motivation for concept-based learning is to go beyond where CRL is not applicable, but still would find it compelling if there could be a setup where the advantage (in terms of the fewer number of domains required) could be showcased.
> >
> >
> > > Could the reviewer please clarify which specific work they mean? We're more than happy to comment further and have a proper discussion on this in the paper as well.
> >
> > The works that I can recall are: https://proceedings.mlr.press/v238/ahuja24a/ahuja24a.pdf, https://arxiv.org/pdf/2311.12267, https://arxiv.org/pdf/2406.05937 (there are probably more)
> >
> > Thanks again for your rebuttal. I like this work and its direction and will maintain my score for now and adjust it if needed after discussion with other reviewers.

---

> ### Author Response · Authors · 2024-08-12
>
> > The works that I can recall are: https://proceedings.mlr.press/v238/ahuja24a/ahuja24a.pdf, https://arxiv.org/pdf/2311.12267, https://arxiv.org/pdf/2406.05937 (there are probably more)
>
> Thanks for clarifying. We have taken another look at these papers, and to the best of our knowledge, these papers still require a number of environments that is lower bounded by the dimension of the latent space $d_z$ (with few notable exceptions with purely observational data, on which we comment below, please see also the footnote in page 3 where we state this). Thus, these papers do not seem to support the claim that there are "CRL methods that guarantee weaker identification requiring _much less interventions_". If we are mistaken, and the reviewer can pinpoint a specific result in one of these papers that accomplishes this, we would be happy to discuss further.
>
> It is true that the work https://proceedings.mlr.press/v238/ahuja24a/ahuja24a.pdf (reference [2] in the paper) and similarly [54, 40] show identifiability from just a single environment as we state in the footnote in the paper. Note, however, that they all make restrictive assumptions on the mixing function and on the distribution of all variables of the latent space, e.g., in the case of [2] it is assumed that the mixing function is polynomial and the support of the latent variables is the Cartesian product of bounded intervals.
> Therefore, neither these results nor their techniques extend to general mixing functions
> or settings where we only make assumptions on the distribution on some of the latent variables. This contrasts with our work, where we make minimal assumptions on the mixing function and only make assumptions on the latent distributions with respect to the concepts of interest.
>
> > However, since the paper contrasts itself a number of times to CRL requiring many interventions, I don't see it totally beyond the scope of this work to make a minimal effort at contrasting to basic CRL methods. I agree with the authors that the motivation for concept-based learning is to go beyond where CRL is not applicable, but still would find it compelling if there could be a setup where the advantage (in terms of the fewer number of domains required) could be showcased.
>
> Thank you for the question. Making an apples-to-apples comparison to CRL is not straightforward because the goals are different. However, we can say the following:
> - Experimentally, we would like to clarify that in cases where _CRL is also applicable_, our algorithm, which is similar to Buchholz et al. [13], can always be applied. Indeed we clarify in the paper that our algorithm is inspired by their works, please see L370-371, 1451.
> However, in the case of sublinear number of environments in our setting, standard CRL techniques are not applicable whereas our work is applicable, and this is the main motivation for our paper.
> - Theoretically, if we can intervene as per the concept distribution we work with (as also described in L129-134), then our results show that $2n$ concept interventions suffice to learn $n\ll d_z$ concepts, and existing methods would not handle this setting to the best of our knowledge (please see also Appendix C for alternate technicalities). Thus, in our setting, current CRL results require $\Omega(d_z)$ interventions, whereas we only need $o(d_z)$ interventions.
>
> Of course, part of our contribution is to propose a different goal (learning concepts) under different assumptions (conditioning vs. intervention), which makes such comparisons difficult. Nonetheless, we hope this offers some insight to understand how our work compares.

---

> > ### Comment · Reviewer_jSX2 · 2024-08-13
> >
> > Thanks again to the authors for their helpful clarifications and explanations.
> >
> > If I remember correctly, the results in [2] apply to general diffeomorphisms, and the number of required environments grows slower than $\Omega(d_z)$, however, I totally agree that the goals are a bit different and that is why I liked the paper in the first place. Relaxing the uncovering of the unmixing for the identification of concepts is quite nice, and I agree (with the help of the author's clarification) that an apples-to-apples comparison might not be possible or fair.
> >
> > Thanks again, and I'm optimistic that the authors will take the various feedback from all the reviewers into account to update the manuscript and the presentation, and I'd like to see this work accepted, therefore, I'm raising my score.

---

### Official Review · Reviewer_KRMD · 2024-07-29

**Soundness:** 3
**Presentation:** 4
**Contribution:** 3
**Rating:** 6
**Confidence:** 3

**Summary:**

The authors argue the shift from causal representation learning (CRL) to concept-based representation learning since the current CRL framework relies on strong requirements such as interventional datasets and stands far from realistic, practical use-cases. The paper formalizes the notion of concepts and establishes a theoretical foundation on the identifiability of concepts. The experiments demonstrate the utility of the framework.

**Strengths:**

- The motivation is convincing and the framework is novel. It also provides a rigorous foundation for the notion of concept and its identifiability.
- The writing is clear and easy to follow. The paper provides a thorough literature review which makes it very helpful to understand the paper positioning and key contributions.
- Experimental results on CLIP and LLMs are interesting. It supports the paper’s motivation to move from CRL to concept-based representation learning.

**Weaknesses:**

Datasets $X^e$ from each environment is associated with different concept $C^e$ and corresponding valuation $b^e$. The proposed method using contrastive learning requires how the dataset is partitioned into each environment, i.e., $X^0, \cdots, X^m$. This implies that the framework is naturally more useful for **discrete** concepts (i.e., discrete valuation), as showcased in the experiments where the authors use discrete labels. However, as the motivation suggests, concepts could be continuous in many cases (e.g., intensity of the color). Therefore, I have doubt on the practical utility of the proposed framework since it cannot handle such continuous concepts. In other words, the requirement of data partition $X^0, \cdots, X^m$ goes against the motivation of the proposed framework of handling continuous concept valuations.

**Questions:**

- (line 208) Can you elaborate? I mean isn’t $A$ a projector matrix?
- Is there any way to quantitatively measure concepts learned by two different models are how much linearly-related to each other?
- The proposed method using contrastive learning should be described in the main section in more detail. Currently, it appears at the appendix, but I think that the algorithm is a key part of the paper which illustrate the practical utility of the proposed framework.

(minor)

- (line 1171-1172) Parenthesis is not closed.
- (line 370, 1491) “the number of concepts $m$” should be “the number of environments $m$”

**Limitations:**

.

---

> ### Author Rebuttal · Authors · 2024-08-07
>
> We thank the reviewer for their positive review and are glad they find the paper novel, well-written and rigorous. We address the weakness below.
>
> > In other words, the requirement of data partition $X^0, \ldots, X^m$ goes against the motivation of the proposed framework of handling continuous concept valuations.
>
> Apologies for the confusion, but we would like to clarify that our setting indeed handles continuous concepts (as claimed) as follows.
> The actual valuations could be continuous (e.g. we allow Gaussian noise) and our identifiability results do hold in such settings.
> In other words, the concept conditional distribution do not condition on a fixed value but rather allow for noise.
> Let us consider this using the example of intensity which was brought up in the review.
> Assume that we have datasets consisting of images taken at different times of the day. Then the intensity of the colors will fluctuate within each dataset due to slight variations of the time or weather conditions, but they will fluctuate around different mean valuations for each dataset. This matches our assumptions for this concept with continuous valuations.
>
> We also note that the experiments we conduct also involve some continuous valuations, since the latent points lie in only an approximate hyperplane.
> However we acknowledge that our methods may have not been fully probed in complex non-discrete settings and we leave this exciting direction for future work.
>
> We now address the questions raised.
>
> > (line 208) Can you elaborate? I mean isn’t $A$ a projector matrix?
>
> The formal definition 1 allows $A$ to be any linear transformation (e.g. it can be scaled), but as the reviewer noticed, there is no loss in generality in assuming it's a projector matrix, and we choose this definition for technical convenience.
>
> > Is there any way to quantitatively measure concepts learned by two different models are how much linearly-related to each other?
>
> Yes, one option are the $R^2$ metric and the Mean Correlation Coefficient (L373-375) that we report in our synthetic experiments (while we compute them against the ground truth since we know it in the case of synthetic data, these can also be computed across models).
> Indeed, in our CLIP experiments, we use the correlation coefficient to measure to what degree the learned concepts for two different models are linearly related (L1177-1188).
>
> > The proposed method using contrastive learning should be described in the main section in more detail. Currently, it appears at the appendix, but I think that the algorithm is a key part of the paper which illustrate the practical utility of the proposed framework.
>
> We appreciate the reviewer's acknowledgment of the contrastive learning method. However, we have regrettably deferred the details to Appendix F due to lack of space. With the additional page available in the final version, we are happy to include this in the main paper.
>
> We thank the reviewer for the additional typographic suggestions and welcome additional feedback to improve the paper.

---

### Author Rebuttal · Authors · 2024-08-07

We appreciate the thoughtful reviews and suggestions by the reviewers.
We are glad that the reviewers found our approach original and novel (reviewers KRMD, jSX2, kVe2), well-written and rigorous (reviewers ohha, KRMD) and significant (reviewers jSX2, ohha).
The reviewers appreciated our theoretical results and its significance, e.g. "I find the theoretical result insightful and important ... also clearly demonstrates the theoretical advantage of concepts vs. interventions", ""The paper is of high quality, providing new important results for identifiability of latent concepts"".
We first address repeated comments below.

**On additional experiments:**
We appreciate the interesting suggestions to extend our method to semi-synthetic datasets and other real-world datasets. Following the suggestion of reviewer jSX2, we have also scaled up our synthetic experiments, please see attached pdf.
However the main contribution of this work is theoretical.
Indeed, not all of the theory community is aware of the empirical support for linearity of representations, and this is a key motivation for our work.
We leave it to future work to comprehensively study experimental methods towards concept learning via our framework.

**On relation between theory and experiment:**
Our synthetic experiments serve to verify the theory whereas the other experiments are a bit more exploratory and serve to probe the different assumptions and conclusions of our theory (see also the response to reviewer ohha).

In the individual responses, we have addressed the weaknesses and answered the questions raised by the reviewers.

---

### Comment · Area_Chair_mF3X · 2024-08-08
**Author-Reviewer Discussion Phase (ends Aug 13 midnight AOE)**

Dear reviewers,

Thank you for your efforts so far! The authors have followed up with each of you to respond to your reviews and have also submitted an overall response (which includes a table of experimental evaluations in the PDF). Could you follow up with them to ask any clarifying questions that may further inform your opinion on the paper?

With thanks, AC

---

### Decision · Program_Chairs · 2024-09-25

**Decision:**

Accept (poster)

**Comment:**

In this submission the authors argue for a shift from causal approaches to representation learning to concept-based approaches. One of the key questions that the authors are interested in is whether such representations are identifiable. Instead of learning causal representations, where we require access to interventional distributions, the authors assume access to a set of concept-conditional distributions. They then define a geometric notion of concepts as linear subspaces of the laten space and prove that these subspaces are identifiable under the assumption of linear independence of concepts, as well as some assumptions about data diversity. The authors evaluate the proposed framework in the context synthetic experiments, multi-modal CLIP models, and LLMs.

After discussion, all reviewers support acceptance and overall this submission appears above the bar. Some comments from reviewers that should be addressed in the camera ready include moving the algorithm block from the appendix to the main text, addressing some points where terminology could be introduced more carefully and providing a greater degree of intuition behind some of the mathematical assumptions. The AC would also recommend that the authors revise their presentation of the experiments to address some of the questions raised by reviewer ohha regarding the connection between experiments and the developed theory.